# Three-Dimensional Hepatocyte Spheroids: Model for Assessing Chemotherapy in Hepatocellular Carcinoma

**DOI:** 10.3390/biomedicines12061200

**Published:** 2024-05-28

**Authors:** Felix Royo, Clara Garcia-Vallicrosa, Maria Azparren-Angulo, Guillermo Bordanaba-Florit, Silvia Lopez-Sarrio, Juan Manuel Falcon-Perez

**Affiliations:** 1Exosomes Laboratory and Metabolomics Platform, Center for Cooperative Research in Biosciences (CIC bioGUNE), Basque Research and Technology Alliance (BRTA), 48160 Derio, Spain; cgarcia@cicbiogune.es (C.G.-V.); maria.azparre.2e@gmail.com (M.A.-A.); guillermo.bordanabaflorit@bio-bizkaia.eus (G.B.-F.); slopez@cicbiogune.es (S.L.-S.); 2Centro de Investigación Biomédica en Red de Enfermedades Hepáticas y Digestivas (CIBERehd), 28029 Madrid, Spain; 3IKERBASQUE, Basque Foundation for Science, 48013 Bilbao, Spain

**Keywords:** primary hepatocyte, extracellular vesicles, sorafenib, dacarbazine, methotrexate

## Abstract

Background: Three-dimensional cellular models provide a more comprehensive representation of in vivo cell properties, encompassing physiological characteristics and drug susceptibility. Methods: Primary hepatocytes were seeded in ultra-low attachment plates to form spheroids, with or without tumoral cells. Spheroid structure, cell proliferation, and apoptosis were analyzed using histological staining techniques. In addition, extracellular vesicles were isolated from conditioned media by differential ultracentrifugation. Spheroids were exposed to cytotoxic drugs, and both spheroid growth and cell death were measured by microscopic imaging and flow cytometry with vital staining, respectively. Results: Concerning spheroid structure, an active outer layer forms a boundary with the media, while the inner core comprises a mass of cell debris. Hepatocyte-formed spheroids release vesicles into the extracellular media, and a decrease in the concentration of vesicles in the culture media can be observed over time. When co-cultured with tumoral cells, a distinct distribution pattern emerges over the primary hepatocytes, resulting in different spheroid conformations. Tumoral cell growth was compromised upon antitumoral drug challenges. Conclusions: Treatment of mixed spheroids with different cytotoxic drugs enables the characterization of drug effects on both hepatocytes and tumoral cells, determining drug specificity effects on these cell types.

## 1. Introduction

For some time now, the recognition that 2D monolayer cells on plastic plates do not fully represent various aspects of the in vivo environment has prompted the development of better hepatocyte models for drug screening [1]. Recognizing the critical role of the cellular environment in shaping cell behavior, significant efforts have been made to recreate the stem cell niche in laboratory settings, aiming for more precise physiological reproduction of spatiotemporal cell–cell interactions, understanding this concept as the evolving relationships between various cell types within the structure, including their spatial organization, as the culture evolves [2]. Pursuing this objective, there is a growing trend towards recreating in vitro as much of the tissue architecture and function observed in vivo as possible. However, demonstrating the physiological relevance of increasingly complex models presents considerable challenges. For tumoral models, one promising approach is to recapitulate the microenvironment surrounding tumors, including vascularization, stromal cells, immune cells, cancer-associated fibroblasts, and microvascular cells, in addition to cancer epithelial cells [3]. Recreation of vascularization has been achieved in tumoral models by combining a bioprinting strategy of human umbilical vein endothelial cells with multicellular spheroids derived from glioma cells [4]. Another aspect is the compositional material of the microenvironment, which can be achieved with multi-material bioprinting [5]. While bioengineers may aim to develop in vitro models that replicate specific tissue features relevant to physiological or diseased functions, a pragmatic perspective also supports simpler models for the majority of users. In this context, models featuring one or two types of cells in 3D culture models are often more robust for mechanistic studies and applications than their more intricate counterparts [6].

One of the models that combine effectiveness with simplicity is the spheroid cell culture [7]. In the absence of surface adhesion, spheroid formation is a forced phenomenon dependent on adhesion molecules like integrins [8]. Cell spheroids exhibit a structure with elongated cells on the surface and layers of cells inside, comprising both growing and non-growing cells, as well as cells experiencing low oxygen levels [9]. An important concept is to distinguish between cell aggregates and spheroids, as the latter involves specific cell–cell interactions within a structured 3D framework and consistent geometry, leading to the development of pathophysiological gradients within the spheroid, which are largely influenced by their diameter [10]. Spheroids smaller than 150 µm display 3D cell–cell and cell–matrix interactions, yet distinct radial proliferative and pathophysiological gradients may not yet be fully evident, typically observed in spheroids ranging from 200 to 500 µm [10]. The 3D spheroid culture of primary hepatocytes, without the addition of extracellular matrices or scaffolds, has been characterized for various pharmacological applications [11,12]. When compared to 2D monolayer or sandwich-cultured primary hepatocytes, these spheroids maintain a more physiologic phenotype of native hepatic tissue at both protein and RNA levels, as evidenced in hepatotoxicity assays [13]. In addition, they exhibit long-term viability over several weeks, retaining physiological features [14]. Markers specific to hepatocytes, such as ALB, HNF4A, and MRP4, can be assessed, along with some of the cytochromes for drug metabolization such as CYP3A4 (whose homolog in mice is CYP3A11) [15]. 

As necessary as modeling the physiological status of the tissue is to find good models of pathological status. The discovery of therapies and biomarkers for hepatocellular carcinoma (HCC) requires suitable in vivo and in vitro models that accurately reflect the biology and heterogeneity observed in patients. While extensively used, 2D-grown cancer cell lines have limitations, particularly as HCCs exhibit significant genetic and phenotypic heterogeneity within tumors and across patients [16]. Another approach involves the use of patient-derived xenograft models, where HCC tissue is transplanted into immunodeficient mice. These models provide valuable insights into histological features and tumor-stroma interactions. However, they are resource-intensive and not well-suited for high-throughput screening [17]. For that reason, an alternative to in vivo models is 3D cultures, which have also been employed for drug screening, finding higher resistance in 3D models as compared to 2D [18]. For analogous reasons as previously discussed, spheroids present a promising model due to their potential for a diverse mix of cell types, making them more similar to solid tumors than 2D cell cultures [19]. Consequently, they offer improved potential for studying the distinctions between tumor cells and healthy cells. They are particularly useful for developing clinical treatments and monitoring how cells respond to drugs [20,21]. Models of simple or co-culture printing of a co-patterned hepatoma and glioma cell-based model were employed to observe the anticancer activity of the drug compound on tumoral cells [22]. Moreover, multicellular tumor spheroids exhibit greater chemotherapeutic resistance compared to the same cells in monolayer culture, a phenomenon that has been described as influenced by adhesion molecules. Changes in the expression of these molecules correlate with altered chemotherapeutic resistance [23]. Additionally, spheroid formation is mediated by integrin-mediated adhesion, which is also associated with chemoresistance to paclitaxel [24].

In the current study, the main goal is to generate reproducible spheroids, as a 3D culture model potentially suitable for automation. This model allows for maintaining primary hepatocytes closer to their differentiated state and for co-culture with hepatocarcinoma cells. Through this approach, the specific effects of antitumoral treatments could be investigated in a more physiological manner compared to 2D mono-cultures. Additionally, it allows for the simultaneous study of the effects on both primary and tumoral cells. While this configuration is ideal for high-throughput screening, specific assays for spheroid analysis are scarce and often require customization, presenting a significant challenge [9]. For that reason, in the present work, we have also explored more efficient means of characterizing the processes associated with the spheroid status, such as the study of extracellular vesicles (EVs). It has already been observed that EVs are released by spheroids into the media [25] and that can be analyzed in the cell culture media without disrupting the spheroid structure [26]. EVs are small membranous bodies that carry proteins, enzymes, nucleic acids, and lipids, and they reflect the physiological state of the secreting cells [27]. This feature could serve as a valuable tool for monitoring the spheroid’s progression during treatments.

## 2. Materials and Methods

### 2.1. Animals and Liver Perfusion

Primary hepatocytes were obtained from the strain resulting from the cross of Albumin-Cre (B6.Cg-Speer6-ps1Tg(Alb-cre)21Mgn/J) and B6.129(Cg)-Gt(ROSA)26Sortm4(ACTB-tdTomato,-EGFP)Luo/J (or mT/mG), both purchased from the Jackson Laboratory. The mT/mG mouse serves as a double-fluorescent Cre reporter, expressing membrane-targeted tandem dimer Tomato (mT) before Cre-mediated excision and membrane-targeted green fluorescence protein (GFP) (mG) after excision [28]. The B6J inbred mouse background is among the most frequently used strains due to its stability and ease of breeding. Many transgenic mice, including those generated using the Cre-lox system, such as the one employed in this study, are produced with a B6J background [29]. Upon crossing with an animal expressing the CRE protein under the Albumin promoter, we obtain an animal with GFP targeted to membranes in all cells expressing albumin, such as hepatocytes. This allows us to distinguish hepatocytes from other cells present in the cultures, such as tumoral cells, and also enables us to track hepatocyte-derived EVs in the culture media. Throughout the article, we will refer to this F1 as TMCRE.

Both procedures and husbandry were conducted in accordance with the Spanish Guide for the Care and Use of Laboratory Animals (RD 53/2013—BOE-A-2013-1337) and regional under Basque Country ethical committee approval (P-CBG-CBBA-0219). Animals between 9 and 12 weeks were subject to a non-recovery surgery performed under anesthetic inhalation of isoflurane (IsoFLO, Chicago, IL, USA) in pure oxygen as the carrier gas (5% for induction, 2–3% for maintenance) and a subsequent two-step collagenase technique which involves sequential perfusion of the liver, as described in [30]. Briefly, a laparotomy was performed to expose the liver, portal vein, and infrahepatic inferior vena cava. The inferior cava vein was cannulated to ensure continuous perfusion at a rate of 1.7 mL/min with 20 mL of Leffert buffer (Leffert buffer: 4-(2-hydroxyethyl)-1-piperazineethanesulfonic acid (HEPES) 10 mM, KCl 3 mM, NaCl 130 mM, NaH_2_PO_4_ 1 mM, glucose 10 mM) with 250 mM ethylene glycol bis(β-aminoethyl ether)-N,N,N’,N’-tetraacetic acid (EGTA). To enable adequate outflow, the portal vein was opened. Subsequently, at the same rate, 20 mL of Leffert buffer was perfused, followed by 25 mL of Leffert buffer perfused with 7500 units of collagenase type I (Worthington Biochemical Corp, Lakewood, NJ, USA) and 110 mM CaCl_2_ at a serial rate change. Following the removal of catheters, the liver was manually sliced and explanted into a Petri dish containing Dulbecco’s Modified Eagle Medium (DMEM) from GIBCO, (Thermo Fisher Scientific, Waltham, MA USA). Finally, the DMEM containing the liver cells was filtered through a cell strainer into 50 mL Falcon tubes. Hepatocytes were pelleted in cold DMEM media at 40× *g* for 4 min. The resulting cell pellet was resuspended in 50 mL of 27% Percoll (GE Healthcare, Chicago, Il, USA, ref#17-0891-01) in DMEM. The tube was gently inverted and centrifuged at 50× *g* for 10 min for the density separation. The resulting cell pellet was washed with 30 mL of cold DMEM and centrifuged at 50× *g* for 5 min. This step was repeated twice, and the final pellet was resuspended in 20 mL of DMEM for counting and subsequent seeding. Only those preparations with vitality over 70%, according to the staining with Trypan Blue Solution 0.4% (Thermo Fisher Scientific, Waltham, MA, USA), were employed for subsequent procedures.

### 2.2. Cell Culture and Spheroid Formation

Spheroids were formed in Nunclon Sphera 3D Plates with Low Attachment and U-Bottom 96-well Plates (Nunc, (Thermo Fisher Scientific, Waltham, MA, USA)) by seeding 5000 vital hepatocytes per well (negative for staining with Trypan Blue Solution 0.4%) in a 200 µL volume of a specific media prepared in Advance DMEM/F12 (GIBCO, (Thermo Fisher Scientific, Waltham, MA, USA) with HEPES, GlutaMax, and Penicillin–Streptomycin, B27 (minus vitamin A) (GIBCO, (Thermo Fisher Scientific, Waltham, MA, USA), plus 15% of RSPO1 conditioned medium (HA-R-Spondin 1-Fc 293T Cells, Cultrex™, Minneapolis, MN, USA), 50 ng/mL EGF (Peprotech), 1.25 mM N-acetylcysteine (Merck KGaA, Darmstadt, Germany), 10 nM gastrin (Tocris, Bristol, UK), 3 mM CHIR99021 (Tocris, Bristol, UK), 25 ng/mL HGF (Peprotech™,Thermo Fisher Scientific, Waltham, MA, USA), 50 ng/mL FGF7 (Peprotech, Thermo Fisher Scientific, Waltham, MA, USA), 50 ng/mL FGF10 (Peprotech, Thermo Fisher Scientific, Waltham, MA, USA), 1 mM A83-01 (Tocris, Bristol, UK), 10 mM Nicotinamide (Merck KGaA, Darmstadt, Germany), and 10 mM Rho Inhibitor g-27632 (Stemcell Technologies, Vancuver, Canada), Primocin (InvivoGen, San Diego, CA, USA) [15], and the medium was refreshed at most every three days by replacing 40% of the volume to ensure spheroid integrity. The duration of the cultures varied according to the experiments, ranging from 7 to 21 days, and is described for each study in the text of the Results section and in the figure legend. When cultured in 2D, hepatocytes were maintained in collagen-coated culture plates at a density of 10 million cells in a 150 mm petri dish for the indicated time.

Tumoral cell lines HepG2 (ATCC^®^ No. HB-8065), SK-HEP1 (ATCC^®^ No. HTB-52™), and HUH7 (No. JCRB0403) were cultured for maintenance in DMEM media supplemented with 10% FBS (Thermo Fisher Scientific, Waltham, MA, USA) and an antibiotic mixture (penicillin, streptomycin, and amphotericin (Thermo Fisher Scientific, Waltham, MA, USA)) in a humidified incubator at 37 °C with 5% CO_2_. To culture mixed spheroids, only tumoral cells at passages below 10 subcultures were employed. To form mixed spheroids, cells were trypsinized and counted; then, 500 cells were added and mixed simultaneously with 5000 primary mouse hepatocytes per well (ratio 1:10 of tumoral vs. non-tumoral) in Nunclon Sphera 3D Plates and using the same specific media described above. In experiments where tumoral cells displayed red fluorescence, the tumoral cells were stained with Vybrant™ Dil Cell-Labeling Solution (Thermo Fisher Scientific, Waltham, MA, USA) following the manufacturer’s protocols. For cytotoxic drug treatment, the compound was added to the spheroid growth media after day 4 of spheroid formation and refreshed at the same time as the control. The drugs employed and their concentrations were sorafenib (Raybiotech, Peachtree Corners, GA USA) at 50 µM [31], dacarbazine at 50 µM (Raybiotech, Peachtree Corners, GA USA) [32], and methotrexate (Merck KGaA, Darmstadt, Germany) at 50 µM [33].

### 2.3. Spheroid Imaging

In vivo images were captured directly using a tissue culture inverted Axiovert fluorescent microscope (Leica, Wetzlar, Alemania). Later, spheroids were manually measured using Image J software (1.53 t) by determining the diameter or calculating an average of the long and short diameters. 3D reconstructions were created by placing spheroids on a carved slide and mounting them directly in Fluoromount-G with DAPI (Thermo Fisher Scientific, Waltham, MA, USA). Confocal images were taken at 1 µm intervals up to 35 µm using a Leica TCS SP8 confocal microscope. 

For histological analysis, spheroids were washed in PBS, fixed with 4% paraformaldehyde in PBS (Merck KGaA, Darmstadt, Germany) for 1 h, and then embedded in pre-liquified HistoGel™ (Thermo Fisher Scientific, Waltham, MA, USA) following the manufacturer’s instructions. After solidification on ice, they were fixed again in 4% paraformaldehyde and transferred to 50% ethanol for at least 24 h before being embedded in paraffin using an automated tissue processor. Sections were cut with a HistoCore MULTICUT microtome or cryostat (Leica, Wetzlar, Germany), deparaffinized with Histo-Clear I solution (Electron Microscopy Sciences, Hatfield, PA, USA), hydrated through decreasing concentrations of alcohol solutions, and stained with hematoxylin–eosin.

For proliferation and apoptosis immunohistochemistry assays, slides underwent antigen retrieval in 10 mM sodium citrate buffer (pH 6.0). The sections were incubated in 0.2% Triton-X in PBS containing 1% BSA and 10% horse serum for 1 h, followed by avidin–biotin blockage. For proliferation analysis, the sections were incubated with Ki67 (DAKO M7249), and for apoptosis analysis, the ApopTag^®^ Peroxidase In Situ Apoptosis Detection Kit (Merck KGaA, Darmstadt, Germany) was employed according to the manufacturer’s instructions. Additionally, immunofluorescence imaging was performed by mounting the slides with Fluoromount-G with DAPI (Thermo Fisher Scientific, Waltham, MA, USA) after hydration.

### 2.4. Flow Cytometry

After a 20 min incubation in Tryple (Gibco, Thermo Fisher Scientific, Waltham, MA, USA), spheroids were mechanically disaggregated with a 1000 µL pipette. Cells were washed with eBioscience™ Flow Cytometry Staining Buffer and suspended in a fluid containing a working concentration of SYTOX^®^ Blue Dead Cell Stain (Thermo Fisher Scientific, Waltham, MA, USA). The suspension was immediately passed through a CytoFLEX flow cytometer (Beckman Coulter Ltd., Nyon, Switzerland). For the analysis of fluorescent EVs released into the media, conditioned media were collected, centrifuged at 2000× *g* for 10 min, and then diluted in 0.1 µm filtered PBS at a 1:5 ratio. The diluted samples were analyzed in the CytoFLEX flow cytometer at a flow speed of 10 µL/min, high-performance rate, and UV SCC for gating. Megamix FSC and Megamix SCC beads (Biocytex, Marseille, Francia) were used as references. All recordings were captured for 60 s, after 30 s of stabilization. The device tubing was washed with 0.1 µm filtered PBS between samples.

### 2.5. EV Collection and Characterization

For EV production, spheroid growth media were replaced with exo-free media (DMEM containing 10% ultracentrifuged serum for 16 h at 100,000× *g*) for three days. The media collected from 96 wells were pooled and centrifuged at 2000× *g* for 10 min, followed by 100,000× *g* in a 70Ti rotor (Beckman Coulter Ltd., Nyon, Switzerland) for 90 min. The pellet was washed once in PBS and finally resuspended in 100 µL PBS. Size distribution within EV preparations was analyzed by measuring the rate of Brownian motion using a NanoSight LM10 system (NanoSight, Amesbury, UK), equipped with fast video capture and particle-tracking software. NTA post-acquisition settings were consistent for all samples, and each video was analyzed to provide mean, mode, and median vesicle size, along with an estimate of the concentration.

Cryo-electron microscopy was employed for imaging EV samples. EV preparations were directly adsorbed onto glow-discharged holey carbon grids (QUANTIFOIL, Jena, Germany). The grids were blotted at 95% humidity and rapidly immersed in liquid ethane with the aid of VITROBOT (Maastricht Instruments BV, Maastricht, Netherlands). Images of the vitrified samples were acquired at liquid nitrogen temperature using a JEM-2200FS/CR transmission cryo-electron microscope (JEOL, Tokyo, Japan) operating at an accelerating voltage of 200 kV and equipped with a field emission gun.

### 2.6. Protein Analysis

For Western blotting characterization of cells and EVs, samples were prepared in LDS sample buffer (NuPAGE, Life Technologies), then boiled (5 min at 37 °C; 10 min at 65 °C; 15 min at 90 °C), and finally centrifuged (10,000× *g*, 1 min) before being applied to the gel. Five micrograms of cellular protein were loaded, while less than two micrograms of vesicular protein were loaded. Electrophoresis used 4–12% Bis-Tris Precast gels (NuPAGE, Invitrogen™, Thermo Fisher Scientific, Waltham, MA, USA) and MOPS SDS running buffer (NuPAGE, Invitrogen™, Thermo Fisher Scientific, Waltham, MA, USA), and samples were run for 90 min at 120 V. Proteins were transferred onto polyvinylidene fluoride membranes (Invitrogen™ iBlot™ 2 Transfer Stacks, PVDF, Thermo Fisher Scientific, Waltham, MA, USA) using an iBLOT2 transfer device (Thermo Fisher Scientific, Waltham, MA, USA). Membranes were blocked overnight with primary antibodies in EveryBlot Blocking Buffer (Bio-Rad, Hercules, CA, USA) at 4 °C. Antibodies against selected proteins, according to literature recommendations [34], included Armenian Hamster anti-Cd81 (Eat2) from Serotec, Mouse anti-CoxIV (4D11-B3-E8) from Cell Signaling, Mouse anti-Grp78 (40/BiP) from BD Biosciences, Mouse anti-Hsp90 (68/Hsp90) from BD Biosciences, Mouse anti-Flotillin-1 (610820) from BD Biosciences, and Rb anti-GFP G1544 from Sigma (Merck KGaA, Darmstadt, Germany). The appropriate secondary antibody was incubated in 5% non-fat dry milk for 6 h at 4 °C. Membranes were visualized using the ImageQuant LAS 4000 with Clarity™ Western Enhanced Chemiluminescence Blotting Substrate (Bio-Rad, Hercules, CA, USA).

### 2.7. RNA Analysis

RNA was extracted from a minimum of 32 spheroids using the RNeasy^®^ Micro (Qiagen), which includes a DNase step on the column, and eluted with 15 µL of RNase-free water. Subsequently, complementary DNA (cDNA) was synthesized from 100 ng of eluted RNA using SuperScript VILO Master Mix (Thermo Fisher Scientific, Waltham, MA, USA) in a final volume of 20 µL. The resulting cDNA was diluted 1/10 in Nuclease-free water (Thermo Fisher Scientific, Waltham, MA, USA), and 3 µL was added in a 6 µL reaction with SYBR Green master mix (Applied Biosystems, Thermo Fisher Scientific, Waltham, MA, USA) and 200 nM of primer. The qPCR analysis was performed in a QS5 Applied Biosystems™ thermocycler (Thermo Fisher Scientific, Waltham, MA, USA). The primer sequences for each assayed gene are as follows: *Alb* (F-AGCCCACTGTCTTAGTGAGG, R-TCTTGCACACTTCCTGGTCC), *Hnf4* (F-GCTAAGGCGTGGGTAGGG, R-AGGCTGTTGGATGAATTGAGG), *Cyp3a11* (F-TGGTCAAACGCCTCTCCTTGCTG, R-ACTGGGCCAAAATCCCGCCG), Gck (F-GAGTGCTCAGGATGTTAAG, R-AAGATCATTGGCGGAAAG), *G6pc* (F-CGACTCGCTATCTCCAAGTGA, R-GGGCGTTGTCCAAACAGAAT), *Asgr1* (F-CTGGGTGGAGTATGAAGGCAG, R-GTCAGTTAGGCCAATCCAAGTG), and *Rplp0* (F-CGACCTGGAAGTCCAACTAC, R-ATCTGCTGCATCTGCTTG).

### 2.8. Statistics

To compare between groups, a *t*-test or ANOVA was performed, and in the later case, followed by Tukey HSD post hoc analysis. Analysis was performed using R software.

## 3. Results

### 3.1. Characterization of Mice Hepatocyte Spheroids

Primary hepatocyte cultures, when plated in ultra-low binding plates, spontaneously organize into compact spheroids with well-defined boundaries. Figure 1 illustrates the time-course evolution of spheroids derived from primary hepatocytes obtained through liver perfusion from the TMCRE mice. As mentioned earlier, these mice had hepatocytes that exhibit GFP expression associated with their membrane.

Histological sections of the spheroid on day 7 reveal an acidic mass inside the spheroid, consistent with images of necrosis (Figure 2A). The Apoptag^®^, (Merck KGaA, Darmstadt, Germany) assay indicates areas with highly fragmented DNA (Figure 2B), and there is almost no evidence of cell proliferation (Figure 2C). DAPI staining of the sections shows that most nuclei have disappeared in the center of the spheroid, leading to a mass of dead cells, while cells in the periphery have elongated their morphology to define an interface with the extracellular media (Figure 2D). In addition, some intercellular spaces can also be appreciated in the interior of those spheroids (Figure 2D).

Transcription analysis revealed that after 7 days of culture, there is a reduction in the expression of liver markers compared to primary hepatocytes immediately after liver perfusion (T0). In Figure 3, we present the ddCt values, where zero represents the expression level of T0 relative to the reference gene. Therefore, negative values indicate a decrease in the expression of those genes over time in culture, while positive values are obtained for genes whose expression increases in vitro culture. It can be observed that the expression in spheroids (3D) is more similar to that of freshly isolated hepatocytes compared to 2D cultures for the same duration. However, while the expression of some markers such as *Alb* or *Hnf4* is preserved in spheroids, there is a decrease in enzymes associated with hepatocyte energy metabolism (*G6pc* and *Gck*), as well as a notable drop in *Cyp3a11*. Nevertheless, *Cyp3a11* expression remains more abundant in spheroids compared to hepatocytes cultured in 2D.

### 3.2. Characterization of EVs Released by Spheroids

To analyze EVs in the cell-conditioned media, we switched to exo-free media, and after 72 h, we collected 10 mL for isolation by ultracentrifugation. We successfully identified vesicles from spheroids containing EV markers like Cd81 and Flot1 [34] (Appendix A). Profiling through NTA (Appendix A) and observing typical vesicle morphology via cryo-electron microscopy (Appendix A) confirmed the presence of EVs. Moreover, GFP was also detected in the EV preparations (Appendix A). Taking advantage of the presence of GFP in vesicles, we track EVs directly in spheroids in growth media, using flow cytometry. The adequate gating to analyze EVs was determined by excluding events present in 0.1 μm filtered PBS and using a mixture of fluorescent polystyrene beads as a reference (Appendix A). The expected EV area was validated using EVs obtained from Vybrant DiO-stained cells via ultracentrifugation (Appendix A). TMCRE spheroid media, when acquired under these conditions, showed quantifiable events. Incubating the media with 0.1% Triton X-100 led to a 70% reduction in events (Appendix A). Diluting the media 1:2 resulted in a proportionate reduction in events (Appendix A). Following this procedure to quantify fluorescent events in the spheroid-conditioned media, we observed a time-dependent reduction over culture time (Figure 4, in which “D1,” “D4,” etc., denote the days after seeding).

### 3.3. Characterization of Mixed Spheroids

When tumoral cells were co-cultured together with primary hepatocytes, originally seeded in a ratio of 1:10, they grew assuming different conformations within the spheroid. We explored three tumoral models—namely, SK-HEP1 (with the mixed spheroid named TMCRE_SKHEP1, as employed in the figures), representing a sinusoidal cell line; HepG2 (with the mixed spheroid named TMCRE_HEPG2), derived from hepatoblastoma; and HUH7 (with the mixed spheroid named TMCRE_HUH7), obtained from hepatocarcinoma. Remarkably, despite the co-culture of both primary hepatocytes and tumoral cells, the core of the spheroid consists of primary hepatocytes, as evidenced by Figure 5 and Figure 6, which depict the arrangement of green fluorescent cells at the center of the spheroid, as well as Appendix A, illustrating proliferation at the periphery of the section. Additionally, the presence of tumoral cells within the core cannot be overlooked, particularly in the SK-HEP1 co-culture, where proliferative cells infiltrate and disrupt the integrity of the inner core. It is noteworthy that for TMCRE_SKHEP1 mixed spheroids, numerous tumoral cells appeared uncolored, as the initial labeling became diluted with each cell division, after 7 days of culture. In contrast to primary hepatocytes, tumoral cells exhibited high proliferation, as indicated by the Ki67 proliferation marker (Appendix A). Consequently, mixed (co-culture) spheroids exhibited varying growth rates over time, with SK-HEP1 cell lines growing the fastest and largest. Spheroids comprised solely of hepatocytes showed minimal size changes once formed (Figure 6).

### 3.4. Antitumoral Treatment of Mixed Spheroids

The mixed spheroids underwent treatment with three distinct antitumoral drugs starting from day 4 of their formation. The impact of these drugs on growing cells influenced the size of the spheroid growth, particularly noticeable for the SK-HEP1 cell line (Figure 7 and Appendix A). However, measuring the effect of tumoral drugs through delayed cell growth is complicated for those tumor models with slow growth rates. Therefore, we assessed the proportion of tumoral cells in the spheroids after the treatment, as well as quantified damaged cells using a dead cell stain that fluoresces at 450 nm (Sytox™, Thermo Fisher Scientific, Waltham, MA, USA). Both indicators revealed a reduction in tumoral cells in response to treatment, with Sorafenib showing a more pronounced effect. Notably, the dose employed for the cytotoxic drugs also inflicted damage on hepatocytes (Figure 8). Appendix A illustrates the gating strategy to differentiate tumoral from non-tumoral cells and vital from damaged cells, based on their staining with the vitality reporter. Additionally, we tracked the presence of fluorescent vesicles in conditioned media as an indicator of hepatocyte damage. Interestingly, dacarbazine treatment appeared to decrease the release of vesicles from hepatocytes. In the case of mixed tumoroids with SK-HEP1, the decrease in the percentage of tumoral cells was mirrored by an increase in EVs released by hepatocytes across all drug treatments (Appendix A).

## 4. Discussion and Conclusions

This study aims to characterize spheroids formed by primary mouse hepatocytes as an in vitro tool for multiple screenings. The results demonstrate both the structure of the spheroids and their capability to release extracellular vesicles (EVs). Additionally, their potential for co-culturing with hepatocarcinoma cells to distinguish the effects on tumoral and non-tumoral cells has been investigated. The discussion of each of these elements follows below.

### 4.1. Culture Conditions and Spheroid Formation

When plated in ultra-low affinity U-shape plates, primary hepatocytes self-arranged into compact multicellular 3D structures, featuring an outer proliferative layer of cells and non-proliferating and apoptotic cells in the inner mass, as described in the literature [35]. Employing a scaffold-free method, cells grew together in 3D without requiring a physical matrix. In our study, the observed diameter of primary spheroids falls within the range reported by other authors and appears to be proportional to the number of cells utilized. For instance, spheroids with 1500 cells per well result in a diameter of approximately 200 μm [11], whereas we observed a diameter of 300 μm for our spheroids formed with 5000 cells. Similarly, Jarvinen et al. also formed spheroids using cell numbers ranging from 1500 to 5000 cells, with observed diameters typically falling between 200 and 300 μm [36]. Notably, both studies utilized ultra-low attachment plates for cell seeding, similar to our approach. While it is reported that providing a physical matrix could enhance hepatocellular function [37], spheroids also demonstrated good performance in recapitulating liver function [11]. Although not considered in our study, the degree of compactness of the spheroids will also be an important measure of spheroid activity since cell density reveals the collective variations in the single cells comprised in them, which are connected to growth and cell cycle changes [38].

Another factor of interest could be the media employed. Many studies employed culture media supplemented with dexamethasone, insulin, transferrin, and selenium. This medium is slightly different from the growth factor cocktail employed in our study that should mimic liver regeneration [15]. In our study, we observed a significant decrease in functionality compared to freshly isolated hepatocytes, yet it certainly surpassed the functionality of 2D cultures, consistent with other studies [39]. Notably, literature also reports that spheroids in ultra-low binding plates outperformed those formed using the agarose method. The U-shaped bottom, as well as the coating treatment of the plates, prevents cell spreading and facilitates the formation of single spheroids per well [40]. Strikingly, not all batches of human primary hepatocytes can form spheroids [41], while regarding mouse primary hepatocytes obtained by liver perfusion, all the preparations form spheroids if the cells show viability levels above of 70%. This can also be attributed to the fact that all animal donors had healthy, young livers, maintaining the hepatocytes at their full potential, and cell damage is only associated with circumstances associated with liver perfusion. 

Regarding potential applications for high-throughput analysis, such as drug screening, robustness, and reproducibility are essential. Our findings indicate minimal deviation in spheroid size growth across the 96 wells, which would facilitate automated effect measurements. Furthermore, both our results and those of other studies demonstrate that spheroid production typically occurs within a short timeframe of 5–7 days, ideal for characterization assays [42]. This format looks promising for assessing drug sensitivity and exploring factors related to chemoresistance. Moreover, differences in susceptibility to doxorubicin, everolimus, and cisplatin have been observed in spheroid tumor models compared to monolayer cultures, highlighting variations related to the expression of different molecules and intrinsic characteristics of the spheroid [21]. In addition, it is important to keep in mind that most drugs rely on the liver to break them down or activate them [43]. Furthermore, a crucial aspect of 3D models is to preserve hepatocyte functionality, ensuring accurate assessments of both drug toxicity and chemotherapy efficacy. Even if the entire spheroid is not fully functional like a lobule, hepatocytes with robust functional machinery can metabolize drugs, mitigating toxicity from primary compounds and producing secondary metabolites that are relevant for drug toxicity and chemotherapy testing [44,45].

### 4.2. Release of EVs

Another important aspect is the release of EVs into the media by spheroids. We characterized the vesicles released by hepatocyte spheroids and also observed a decrease in concentration over time, which could serve as a surrogate marker of spheroid maturation. The release of EVs has been studied in various spheroid models, including mesenchymal spheroids, which exhibit distinct properties compared to 2D mesenchymal EVs [46]. In other 3D models, even those embedded in Matrigel, EVs loaded with miRNAs have been reported to be released and taken up by organoid cells, influencing their gene expression [47]. Previous research has also described the release of EVs both into the extracellular media and the inner space of spheroids. In this regard, a protocol involving the hydrolysis of the extracellular matrix deposited by cells confirmed that some EVs are entrapped within the multicellular spheroid [48]. Our observations were limited to the release of extracellular media of a specific fluorescent group of EVs, which we could easily monitor with a flow cytometer. However, we did not monitor the release of EVs from tumor cells, which also contributes to the overall release of vesicles into the conditioned media. To monitor these particles, efficiently expressed constructs of membrane protein-based fluorochromes will be necessary, which may also influence their response to drugs. In future studies, a tracking system for tumoral EVs may inform of the cell damage caused by cytotoxic drugs. In vivo, the increase in liver-derived EVs in the bloodstream has been associated with drug-induced liver damage [49,50,51] or the increase in EV release upon temozolomide treatment in glioblastoma cells [52]. A final point to discuss regarding EVs is the decrease observed in their release upon dacarbazine treatment, a phenomenon lacking explicit explanations or references in the current literature. Dacarbazine-induced cytotoxicity has been previously linked to reactive oxygen species formation and lysosomal membrane leakiness, leading to hepatocyte lysis in isolated rat hepatocytes [53]. However, oxidative stress has been associated with an increase in EV release [54] and do not explain our observation. Therefore, further investigations involving extended time courses and the status of vesicle biogenesis and trafficking would be necessary. 

### 4.3. Tumoral Cells Assayed

Regarding co-culture spheroid, our observations revealed a distinct peripheral distribution of tumoral cells, leaving primary hepatocytes forming an inner core. This specialized distribution of co-cultured cells has been observed previously, where cells displaying epithelial cell markers accommodate themselves in the periphery of the spheroid [55]. The mechanism behind cell distribution requires further analysis, but we have also observed differences between tumoral cell lines. The SK-HEP1 cell line, suspected to derive from an endothelial origin [56], exhibits the fastest growth rate and completely covers the surface of the spheroid. This cell line has metastatic potential, as it has been reported to engraft in nude mice [57]. Additionally, it demonstrates higher resistance to methotrexate compared to other employed cell lines [58]. Interestingly, SK-HEP1 synthesizes various proteins regulating cell attachment, including vimentin [56] and fibronectin [59], with the latter production enhanced by extracellular matrix components [60]. However, it has been described that SK-HEP1 does not exhibit polarization [61], while HUH7 has shown polarity in Matrigel cultures [62], and HepG2 has been described to have a polarization somewhat similar to primary hepatocytes [63]. For the latter, they show typical cell junction proteins such as claudin, e-cadherin, or occluding [64]. However, HepG2 and HUH7 exhibit differences in their integration levels with primary hepatocytes, which are challenging to explain based on their inherent characteristics. Both cell lines exhibit variations in the expression of several protein profiles compared to mature hepatocytes [65], yet they share high similarities in their metabolic characteristics [66]. While their sensitivity to sorafenib in 3D spheroids has been previously described, showcasing similarities in the reduction of cell growth in the presence of the chemotherapy drug [67], their response to collagen presence or environmental stiffness remains consistent between the two [68,69]. HepG2 is also able to form tumors in nude mice at a higher rate than SKHEP1, and it shows less drug inhibition by different chemotherapeutic agents [57]. Notably, metastatic invasiveness could be increased by enriching the population of cancer stem cells through spheroid formation [70]. It is also important to consider that HepG2 presents expression related to fetal development, such as high expression of insulin-like growth factor-1 (IGF-1), which leads to the overactivation of molecular targets associated with the insulin pathway [71]. On the contrary, HUH7 does not exhibit these characteristics regarding insulin pathway activity; however, it has demonstrated a high correlation in gene expression spectrum with drug-metabolizing enzymes, transporters, and glutathione-S-transferase activity compared to primary human hepatocytes [72]. It is interesting to highlight that all three cell lines show typical HCC markers such as AFP, EpCAM, or CD133 [73,74]. Speculatively, we attribute the differences in their aggregation with primary hepatocytes to their distinct origins, with HepG2 being a hepatoblastoma-derived cell line and HUH7 derived from hepatocarcinoma cell lines, showing disparities in certain receptors and responses to fibroblast growth factors 1 or 2 [75].

### 4.4. Chemotherapeutic Agents Used

According to previous studies, the dose we have employed is relatively high [32,33,76], which might also explain the observed effects on hepatocytes. The rationale behind this choice was to achieve a high clearance of tumoral cells from spheroids, in order to obtain clear results regarding the utility of mixed spheroids as a tool. Regarding the drugs employed, sorafenib is a chemotherapy drug used against hepatocarcinoma [77]. Its mechanism of action relies on competitive inhibition of ATP binding to the catalytic domains of various kinases [78], subsequently blocking different signaling pathways such as PDGB (platelet-derived growth factor), vascular endothelial growth factor (VEGF), or Raf signaling [79]. On the other hand, methotrexate interrupts one-carbon metabolism by inhibiting the synthesis of tetrahydrofolate, thereby inhibiting the synthesis of purines and thymidines, impairing cell cycle progression. Cells become resistant by overcoming this limitation, for instance, by overexpressing enzymes of the one-carbon metabolism [80]. Finally, dacarbazine, an alkylating agent commonly used in combination with other chemotherapeutic agents, may act as a purine analog and antimetabolite. Additionally, this drug is extensively metabolized in the liver and produces intermediates, some of which have alkylating activity, causing methylation, modification, and cross-linking of DNA, thereby inhibiting DNA, RNA, and protein synthesis [81]. Contrary to the activation of the drug observed in the liver for dacarbazine, the activity of tyrosine kinase inhibitors like sorafenib has been observed to decrease when CYP3A4 is active [82]. It should be noted that all three drugs can cause liver damage, and the effect on cells is assessable through various analytical approaches, ranging from high-throughput analysis, such as measuring size through automated high-content screening, to flow cytometry analysis for different cell populations. Flow cytometry provides insights into the damage affecting the inner core of hepatocytes induced by cytotoxic drugs. The mechanism of toxicity has been described for all three drugs. Dacarbazine can be metabolically toxic upon its transformation by CYP1A2 [53,83], with the main contribution being the generation of reactive oxygen species [53]. Methotrexate is also converted in the liver, and its derivatives accumulate in different tissues, including the liver, causing damage upon exposure, which is also related to oxidative stress [84,85]. Finally, sorafenib, like other tyrosine kinase inhibitors, has been observed to increase the risk of liver toxicity [86]. The main concern regarding their toxicity is once again the production of reactive metabolites leading to hepatocellular damage through oxidative stress, mitochondrial dysfunction, and impaired glycolysis [87]. As mentioned earlier, the metabolism of sorafenib is primarily catalyzed by CYP3A4 [82]. These dependencies on the hepatocyte machinery justify the application of 3D models, which exhibit a higher capacity for inducible expression of *CYP3A4* compared to 2D cultures [12]. It should be highlighted that, in addition to spheroids, various approaches for complex cultures have been tested. In these cultures, not only primary hepatocytes but also tumor cells may exhibit enhanced functionality. For instance, increased metalloproteinase production was observed in bioprinted HELA cells compared to their 2D counterparts [88]. This underscores the rationale for utilizing 3D methods also to evaluate tumor cell behavior. Future studies, involving a larger number of drugs, could also benefit from this approach to refine the drug dosages. 

### 4.5. Limitations of the Study

We acknowledge that the study had limitations concerning both the cell mixture and the analysis conducted to monitor chemotherapy outcomes. Regarding the cell mixture, many aspects of liver functionality were not replicated, such as the involvement of the immune system and the participation of stellate cells, which could be decisive in tumor progression [89]. The study would also benefit from incorporating a measurement of spheroid mass, which could provide insights into changes in cell density during growth and treatment. Indeed, a parameter of density could reflect modifications in the inner cellular composition and intercellular connection network [38]. However, such measurements require microfluidic devices, wherein single spheroids enter a designed analysis flow channel dedicated to assessing their terminal velocity [90]. With the aid of these specialized devices, values for weight, size, and mass density of the 3D biological sample could be accurately measured. In the same way, the metabolic capabilities of the spheroid and the impact of chemotherapy on both tumoral cells and hepatocytes should be further investigated in detail, including the specific mechanisms of cell death. While tumor spheroids with enhanced functionality have been employed in studies of fundamental cancer biology [91] as well as drug screening and validation [92], they continue to lack the major extracellular matrix component of the tumor microenvironment [3]. Alternative approaches include porous scaffolds, which have shown enhanced proliferation, growth factor and cytokine secretion, and vascularization compared to both 2D monolayer cultures and 3D tissues embedded in Matrigel, both in vitro and after in vivo implantation [93]. Recently, the bioprinting of hepatocarcinoma models using HepG2 and human umbilical vein endothelial cells has emerged as a highly scalable platform for generating engineered tissue constructs containing vascular channels and multiple cell types [5]. Additionally, regarding EV release, it would be valuable to explore changes in the subpopulation of vesicles and the presence of markers of cell death following treatment, for both hepatocytes and tumoral cells. However, despite the limitations listed, the simplicity of the proposed model offers advantages in terms of implementation and result analysis.

## 5. Conclusions

The generation of mixed models involving primary hepatocytes and tumoral cells forming spheroids represents an in vitro approach applicable for studying tumoral properties, sensitivity, and specificity of antitumoral drugs. This model is compatible with various analytical methods for assessing the induced damage in the targeted cells.

## Figures and Tables

**Figure 1 biomedicines-12-01200-f001:**
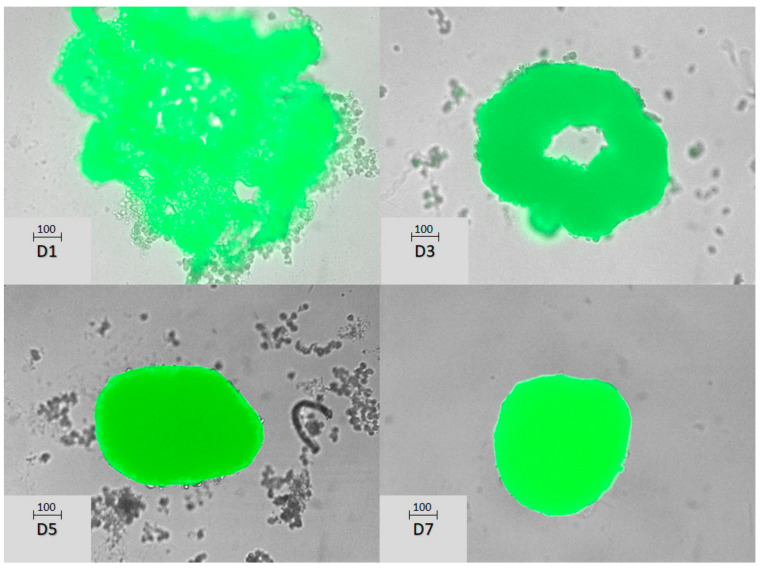
Spheroid formation occurred in hepatocytes from TMCRE mice when 5000 cells were placed per well in ultra-low binding 96-well plates. The scale bars represent 100 µm and images were taken at 1(D1), 3 (D3), 5(D5), and 7(D7) days after cell seeding.

**Figure 2 biomedicines-12-01200-f002:**
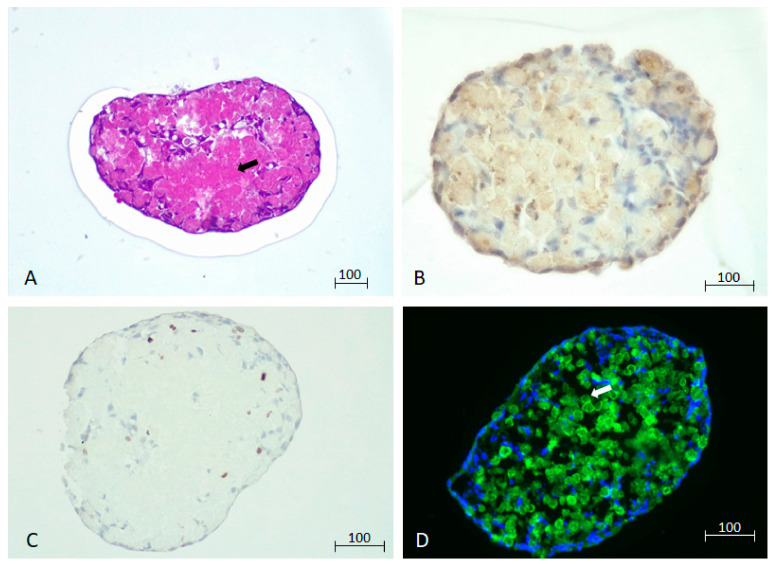
Histological sections of 7-day-old spheroids were stained with (**A**) hematoxylin–eosin or subjected to immunohistochemistry against (**B**) broken DNA fragments (Apoptag^®^, Merck KGaA, Darmstadt, Germany) and (**C**) the cell proliferation marker Ki67 protein. (**D**) DAPI staining reveals nuclei, and the green fluorescence corresponds to the GFP bound to a membrane protein in the hepatocytes. We have indicated with an arrow the inner area, predominantly stained by eosin and lacking nuclei according to DAPI staining, indicative of necrosis. The scale bars represent 100 µm.

**Figure 3 biomedicines-12-01200-f003:**
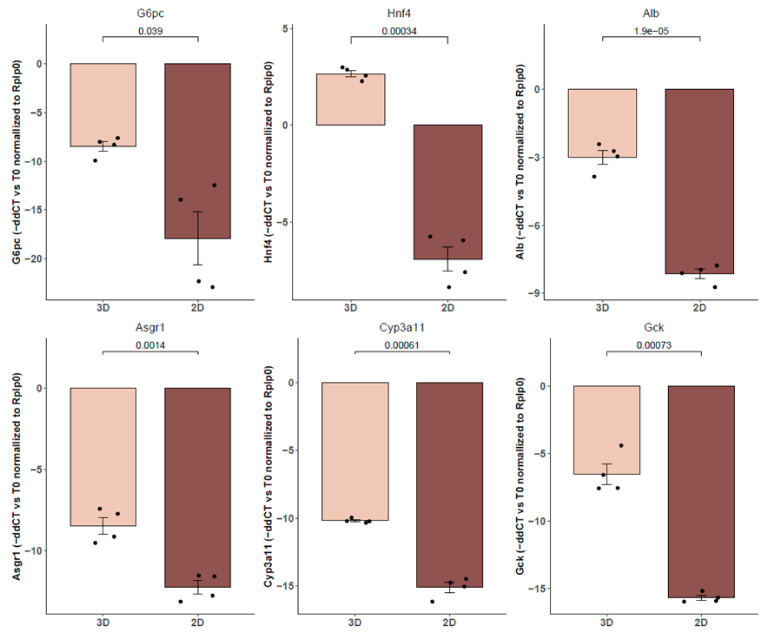
Gene expression of hepatocyte function markers using qPCR. This was achieved for spheroids cultured for 7 days (3D) compared to hepatocytes cultured in a monolayer (2D) for the same time. We normalized the data using ddCt, with freshly isolated hepatocytes (T0) as the reference sample and *Rplp0* as the reference gene. The comparisons were conducted using a *t*-test (n = 4).

**Figure 4 biomedicines-12-01200-f004:**
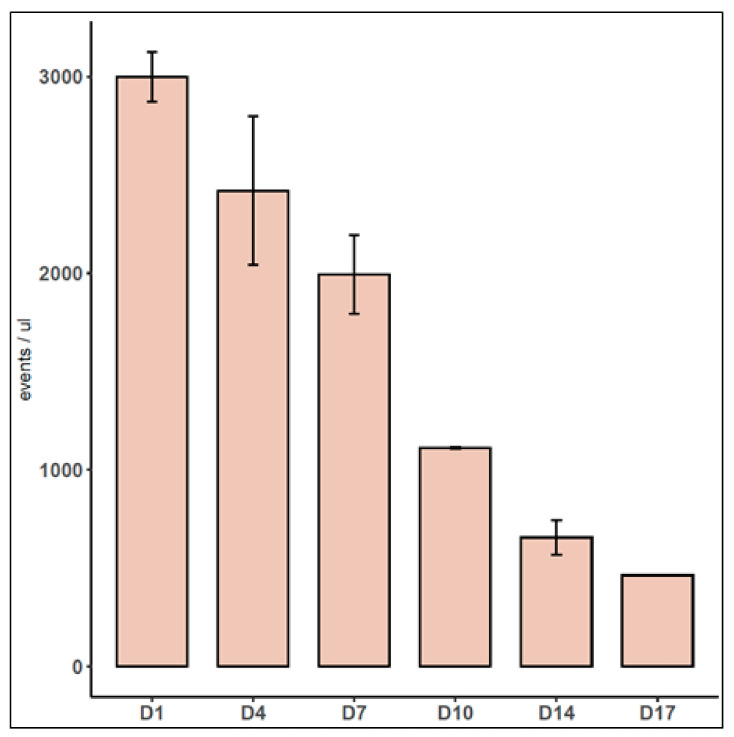
Fluorescence events concentration in the culture media obtained from spheroids along the culture time (n = 3). The cell media were sampled at days 1(D1), 4 (D4), 7(D7), 10 (D10), 14(D14), and 17(D17) after cell seeding.

**Figure 5 biomedicines-12-01200-f005:**
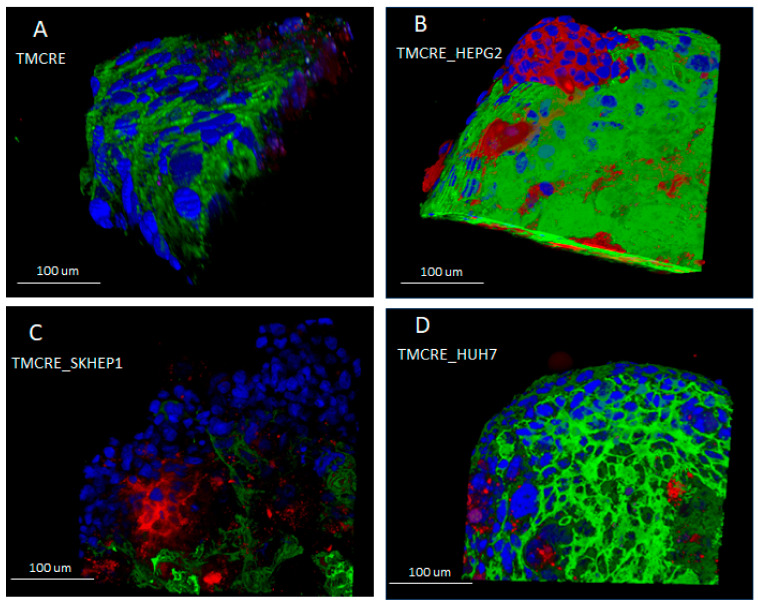
Tridimensional reconstruction of a series of confocal images taken from intact spheroids cultured for 7 days. Panel (**A**) shows the reconstruction of a mono-culture spheroid formed by primary hepatocytes (TMCRE), while panels (**B**–**D**) depict mixed spheroids containing tumoral cells HEPG2 (TMCRE_HEPG2), SKHEP-1 (TMCRE_SKHEP1), and HUH7 (TMCRE_HUH7), respectively. Tumoral cells, labeled before spheroid formation, may appear either reddish or uncolored (see text for details), while hepatocytes exhibit green fluorescence.

**Figure 6 biomedicines-12-01200-f006:**
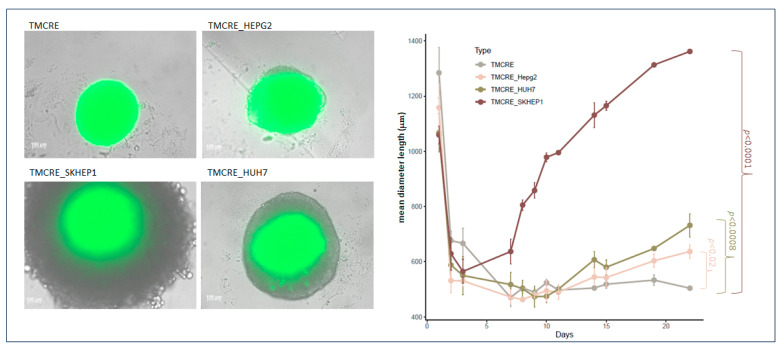
Growth curve of mixed spheroids over time. On the left, an image depicts either a spheroid formed by a mono-culture of primary hepatocytes (TMCRE) or mixed spheroids containing hepatocytes and different tumoral cells; HEPG2 (TMCRE_HEPG2), SKHEP-1 (TMCRE_SKHEP1), and HUH7 (TMCRE_HUH7) at day 22 after seeding. The growth curve was constructed by measuring the mean diameter length (μm) of spheroids over time, highlighting variations in their growth rates compared to hepatocyte spheroids. Statistical analyses, including ANOVA and planned contrasts for each tumoral type versus hepatic control, were conducted (n = 3).

**Figure 7 biomedicines-12-01200-f007:**
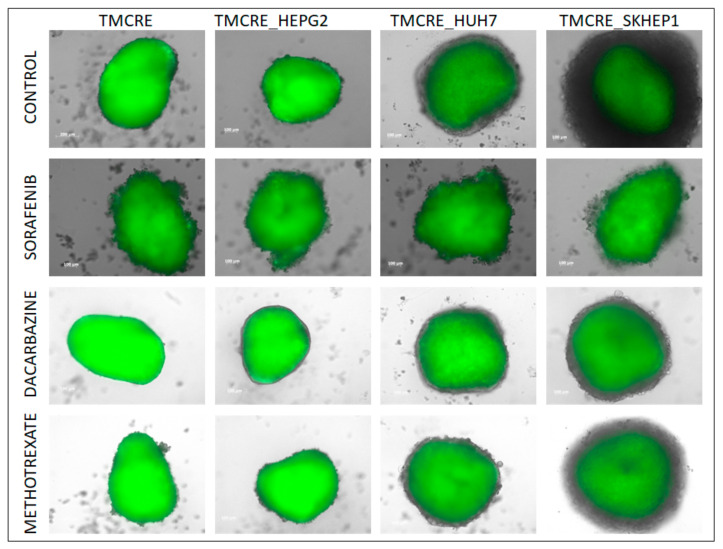
Live imaging at day 14 after seeding of spheroids formed by mono-culture primary hepatocytes (TMCRE) or mixed spheroids obtained by co-culture of hepatocytes and different tumoral cells; HEPG2 (TMCRE_HEPG2), SKHEP-1 (TMCRE_SKHEP1), and HUH7 (TMCRE_HUH7), treated with three different drugs at day 4 after seeding. Green cells are hepatocytes obtained from TMCRE animals, which express GFP bound to a membrane peptide.

**Figure 8 biomedicines-12-01200-f008:**
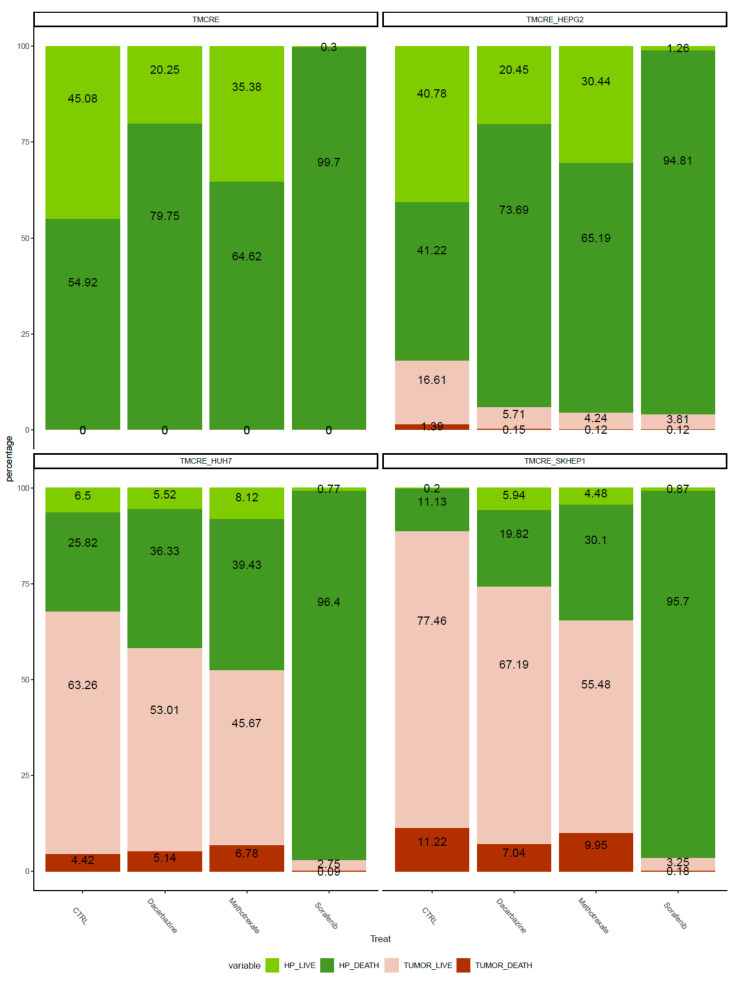
Percentage of damaged hepatocytes and tumoral cells in mono-cultures of primary hepatocytes spheroids (TMCRE) or mixed spheroids containing hepatocytes and different tumoral cells; HEPG2 (TMCRE_HEPG2), SKHEP-1 (TMCRE_SKHEP1), and HUH7 (TMCRE_HUH7). The results are percentages of stained cells according to flow cytometry analysis. In the graph, light green corresponds to vital hepatocytes (HP_LIVE), while dark green indicates damaged hepatocytes (HP_DEATH). Non-green bars represent tumoral cells, with light pink representing viable tumoral cells (TUMOR_LIVE), and the red segment of the bars indicating the percentage of damaged tumoral cells (TUMOR_DEATH). Data were obtained from a pool of 24 spheroids per condition. Treatment started on day 4, and the spheroid collection was taken on day 14 after seeding.

## Data Availability

All the data are available in the Appendix A section.

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
