# Peer review of "Three-Dimensional Hepatocyte Spheroids: Model for Assessing Chemotherapy in Hepatocellular Carcinoma"

_biomedicines, 2024, doi:10.3390/biomedicines12061200_

Round 1

Reviewer 1 Report

Comments and Suggestions for Authors

The manuscript entitled - Three-Dimensional Hepatocyte Spheroids: Model for Assessing Chemotherapy in Hepatocellular Carcinoma-is dealing with The 3D spheroid culture of primary hepatocytes, without the addition of extracellular matrices or scaffolds.

In principle, this topic is interesting in respect to establish simpler but also accurate models in cancer research. The aim and introduction clearly addresses the need for these models in comparison with alternative strategies.

In the Material and Methods part the different methods and techniques are explained. For the statistical evaluation the R-software is used – this should be specified. 

In the Result section the model development and detailed results are shown and several supplementary files are provided for detailed information. In principle the figures are well prepared and suitable.

Minor comments:

Figure 5. Tridimensional reconstruction of confocal imagens of intact mixed spheroids, showing tumoral cells in red or without colour, and hepatocytes in green. 321 – Why in red and without color.

the viability indicator Sytox – What is special on Sytox

In the Discussion the authors first of all discuss the 3D spheroids with the literature. Regarding to that several aspects should be recognized:

-       Size of the spheroids compared to other studies

-       Potential media effects, respectively comparison with other existing 3D models

-       Is only the u-shape of the wells inhibiting cell spreading or are other parameters relevant 

Regarding potential applications for high throughput:

-       What is exactly meant by high throughput

-       It is also important to have hepatocyte functionality - is this ensured although the whole spheroid is not functional

Release of EVs:

-       what is in detail technically needed to distinguish between hepatocyte- and tumor EVs

Toxicity:

-       it´s obvious that all cells are more or less damaged by different tumor agents. Is this model under and under which circumstances accurate for the screening of new therapeutics? How could site effects be distinguished from anti-cancer efficacy and are these estimations in-vivo confirmed.

Comments on the Quality of English Language

English phrasing need to be check, ideally by a native speaker.

Several misspellings are evident, such as were / was; pippete, confocal imagens and many more. Furthermore, inconsistant formatting should be corrected such as xxx°C /xxx °C.

Author Response

Review 1

The manuscript entitled - Three-Dimensional Hepatocyte Spheroids: Model for Assessing Chemotherapy in Hepatocellular Carcinoma-is dealing with The 3D spheroid culture of primary hepatocytes, without the addition of extracellular matrices or scaffolds.

In principle, this topic is interesting in respect to establish simpler but also accurate models in cancer research. The aim and introduction clearly addresses the need for these models in comparison with alternative strategies.

In the Material and Methods part the different methods and techniques are explained. For the statistical evaluation the R-software is used – this should be specified. 

In the Result section the model development and detailed results are shown and several supplementary files are provided for detailed information. In principle the figures are well prepared and suitable.

Firstly, thank you very much for your positive comments and the time you dedicated to the review. In the following paragraphs, we will try to address the questions you raised.

Minor comments:

Figure 5. Tridimensional reconstruction of confocal imagens of intact mixed spheroids, showing tumoral cells in red or without colour, and hepatocytes in green. 321 – Why in red and without color.

As mentioned in the Materials and Methods sections, tumoral cells were stained with the membrane marker Vybrant™ Dil. As cells grow and divide, the amount of label becomes diluted, resulting in some cells showing no color at all in the final image, particularly for SKHEP-1, where proliferation was faster. A sentence has been included in the text to clarify this particular point.

“Additionally, the presence of tumoral cells within the core cannot be overlooked, particularly in the SK-HEP1 co-culture, where proliferative cells infiltrate and disrupt the integrity of the inner core. It is noteworthy that for TMCRE_SKHEP1 mixed spheroids, numerous tumoral cells appeared uncolored, as the initial labelling became diluted with each cell division, after 7 days of culture.”

the viability indicator Sytox – What is special on Sytox

I believe is not a special method. Sytox is just a method to stain cells with a compromised membrane, compatible with flow cytometry. According to the company: “is a high-affinity nucleic acid stain that easily penetrates cells with compromised plasma membranes and yet will not cross the membranes of live cells”.

We rewrite the sentence, so there is no doubt about we are using a commercial method:

“Therefore, we assessed the proportion of tumoral cells in the spheroids after the treatment, as well as quantified damaged cells using a dead cell stain that fluoresces at 450 nm (Sytox™, Thermofisher Scientifics).”

In the Discussion the authors first of all discuss the 3D spheroids with the literature. Regarding to that several aspects should be recognized:

-       Size of the spheroids compared to other studies

-       Potential media effects, respectively comparison with other existing 3D models

-       Is only the u-shape of the wells inhibiting cell spreading or are other parameters relevant 

The observed diameter falls within the same range as other spheroids formed with primary hepatocytes, and formation occurs using different media. A paragraph discussing these aspects has been included in the discussion section. Concerning the plate, certainly is not only the U shape, but rather the ultra-low attachment coating treatment of the plates, which discourages cells from adhering to the plastic and instead promotes their aggregation. We have modified the sentence to clarify those points.

“For instance, spheroids with 1500 cells per well resulting in a diameter of approximately 200 μm [9], whereas we observed a diameter of 300 μm for our spheroids formed with 5000 cells. Similarly, Jarvinen et al. also formed spheroids using cell numbers ranging from 1500 to 5000 cells, with observed diameters typically falling between 200 and 300 μm [36]. Notably, both studies utilized ultra-low attachment plates for cell seeding, simi-lar to our approach.  (…)

The U-shaped bottom, as well as the coating treatment of the plates, prevent cell spreading and facilitates the formation of single spheroids per well [40].”

Regarding potential applications for high throughput:

-       What is exactly meant by high throughput

-       It is also important to have hepatocyte functionality - is this ensured although the whole spheroid is not functional

Certainly, what we miss to mention is that high-throughput analysis typically refers to a cell culture system that allows automation in microplates. An example would be large drug screening, since the lack of a solid or semisolid matrix, could be a benefit to perform some protocols. The sentence has been rewritten to deepen the meaning.

“Regarding potential applications for high throughput analysis, such as drug screening, robustness and reproducibility are essential. Our findings indicate minimal deviation in spheroid size growth across the 96 wells, which would facilitate automated effect measurements. Furthermore, both our results and those of other studies demonstrate that spheroid production typically occurs within a short timeframe of 5–7 days, ideal for characterization assays [42].”

Regarding the hepatocyte functionality, it is crucial to consider various factors, such as drug toxicity, which may rely on the functional machinery of these cells. While the spheroid may not fully replicate the functionality of the hepatic lobule, maintaining hepatocytes as close to their mature state as possible can yield more realistic results. The sentence now read like this:

“Furthermore, a crucial aspect of 3D models is to preserve hepatocyte functionality, ensuring accurate assessments of both drug toxicity and chemotherapy efficacy. Even if the entire spheroid is not fully functional like a lobule, hepatocytes with robust functional machinery can metabolize drugs, mitigating toxicity from primary compounds and producing secondary metabolites that are relevant for drug toxicity and chemotherapy testing [44, 45]”

Release of EVs:

-       what is in detail technically needed to distinguish between hepatocyte- and tumor EVs

To detect EVs derived from tumoral cells using flow cytometry, fluorescent molecules must be attached to the cell membrane. Simple transfection of fluorescent molecules is ineffective because the synthesized fluorochrome might not be secreted through EVs. Alternatively, labeling with specific human tetraspanins carries the risk of falsely detecting antibody complexes. To advance in this area, efficiently expressed constructs of membrane protein-based fluorochromes will be required, as well as evaluated the possible effect of those in the sensibility to chemotherapy. However, our initial attempts revealed that lentivirus transfections were not efficient in some of the cell lines used and consequently, we did not pursue that direction in the present study. The sentence has been modified to include this future line of work.   

“However, we did not monitor the release of EVs from tumor cells, which also contribute to the overall release of vesicles into the conditioned media. To monitor these particles, efficiently expressed constructs of membrane protein-based fluorochromes will be necessary, which may also influence their response to drugs. In future studies, a tracking system for tumoral EVs may inform of the cell damage caused by cytotoxic drugs.”  

Toxicity:

-       it´s obvious that all cells are more or less damaged by different tumor agents. Is this model under and under which circumstances accurate for the screening of new therapeutics? How could site effects be distinguished from anti-cancer efficacy and are these estimations in-vivo confirmed.

Certainly, the model has limitations, and extrapolating the results to clinical settings requires a complex process. The concept is that mixed spheroids may aid in preliminary screening, allowing for the assessment of a large number of drugs in a more physiological context than 2D monocultures. Although the results may be subject to bias, it is also possible to envision precision medicine models where primary tumor cells would be also assessed. A paragraph emphasizing the study's limitations has been appended to the end of the discussion.

“We acknowledge that the study had limitations concerning both the cell mixture and the analysis conducted to monitor chemotherapy outcomes. Regarding the cell mixture, many aspects of liver functionality were not replicated, such as the involvement of the immune system and the participation of stellate cells, which could be decisive in tumor progression [89]. The study would also benefit from incorporating a measurement of spheroid mass, which could provide insights into changes in cell density during growth and treatment. Indeed, a parameter of density could reflect modifications in the inner cellular composition and intercellular connection network [38]. However, such measurements re-quire microfluidic devices, wherein single spheroids enter a designed analysis flow-channel dedicated to assessing their terminal velocity [90]. With the aid of these specialized devices, values for weight, size, and mass density of the 3D biological sample could be accurately measured. In the same way, the metabolic capabilities of the spheroid and the impact of chemotherapy on both tumoral cells and hepatocytes should be further investigated in detail, including the specific mechanisms of cell death. (…)

Additionally, regarding EV release, it would be valuable to explore changes in the sub-population of vesicles and the presence of markers of cell death following treatment, for both hepatocytes and tumoral cells. However, despite the limitations listed, the simplicity of the proposed model offers advantages in terms of implementation and result analysis.”

Comments on the Quality of English Language

English phrasing need to be check, ideally by a native speaker.

Several misspellings are evident, such as were / was; pippete, confocal imagens and many more. Furthermore, inconsistent formatting should be corrected such as xxx°C /xxx °C.

We apologize for the errors in the manuscript. We have reviewed the text and attempted to correct all typos and misspellings.

Reviewer 2 Report

Comments and Suggestions for Authors

This paper describes an investigation of the cytotoxicity of anticancer drugs on mixed spheroid models of liver cancer. The article is scientifically sound and well written. I have just minor comments regarding a few issues of style. 

1. The Introduction of an original research article should provide background information for the reader, argue the importance of the study and identify the knowledge gap addressed by it. The latter is missing from this manuscript. The Introduction of this article is too laconic to present the general context of 3D cancer models. A more complete presentation of the relevant literature is needed in order to point out the original contribution of the present work. In my opinion, the following references might help in this respect. (None of them is mine.) 

[1*] Hirschhaeuser F, Menne H, Dittfeld C, West J, Mueller-Klieser W, Kunz-Schughart LA. Multicellular tumor spheroids: An underestimated tool is catching up again. Journal of Biotechnology. 2010;148(1):3-15.

[2*] Fennema E, Rivron N, Rouwkema J, van Blitterswijk C, de Boer J. Spheroid culture as a tool for creating 3D complex tissues. Trends Biotechnol. 2013;31(2):108-15.

[3*] Zhang YS, Duchamp M, Oklu R, Ellisen LW, Langer R, Khademhosseini A. Bioprinting the Cancer Microenvironment. ACS biomaterials science & engineering. 2016;2(10):1710-21.

[4*] Datta P, Dey M, Ataie Z, Unutmaz D, Ozbolat IT. 3D bioprinting for reconstituting the cancer microenvironment. Precision Oncology. 2020;4:18.

2. In Section 3, the annotations of certain illustrations are incomplete. In Fig. 2, please specify the unit of measurement (micrometers) besides the number written on the scale bar. Alternatively, mention it in the figure caption. In Figures 1 and 4, the abbreviations D0-D14 should be defined in the figure caption. Also, the acronyms used on the individual panels from Figs. 5 and 6 need to be explained in detail, both in the main text and figure caption. Also, it is not clear to me what is meant by "without colour" on line 312. 

3. The Discussion would further benefit from comparisons made with results obtained from different cancer models. For example, 3D bioprinted models have also been used to assess the response of cancer cells to chemotherapy [*5]. 

[5*] Zhao Y, Yao R, Ouyang L, Ding H, Zhang T, Zhang K, et al. Three-dimensional printing of Hela cells for cervical tumor model in vitro. Biofabrication. 2014;6(3):035001.

Author Response

Review 2

his paper describes an investigation of the cytotoxicity of anticancer drugs on mixed spheroid models of liver cancer. The article is scientifically sound and well written. I have just minor comments regarding a few issues of style. 

  1. The Introduction of an original research article should provide background information for the reader, argue the importance of the study and identify the knowledge gap addressed by it. The latter is missing from this manuscript. The Introduction of this article is too laconic to present the general context of 3D cancer models. A more complete presentation of the relevant literature is needed in order to point out the original contribution of the present work. In my opinion, the following references might help in this respect. (None of them is mine.) 

 [1*] Hirschhaeuser F, Menne H, Dittfeld C, West J, Mueller-Klieser W, Kunz-Schughart LA. Multicellular tumor spheroids: An underestimated tool is catching up again. Journal of Biotechnology. 2010;148(1):3-15.

 [2*] Fennema E, Rivron N, Rouwkema J, van Blitterswijk C, de Boer J. Spheroid culture as a tool for creating 3D complex tissues. Trends Biotechnol. 2013;31(2):108-15.

 [3*] Zhang YS, Duchamp M, Oklu R, Ellisen LW, Langer R, Khademhosseini A. Bioprinting the Cancer Microenvironment. ACS biomaterials science & engineering. 2016;2(10):1710-21.

 [4*] Datta P, Dey M, Ataie Z, Unutmaz D, Ozbolat IT. 3D bioprinting for reconstituting the cancer microenvironment. Precision Oncology. 2020;4:18.

We thank the referee for their efforts and the valuable improvements they suggested. The articles recommended are four comprehensive reviews, from which we have tried to incorporate key concepts to highlight the objectives of our current study. Various references have been included throughout the introduction to improve its precision and clarity.

  1. In Section 3, the annotations of certain illustrations are incomplete. In Fig. 2, please specify the unit of measurement (micrometers) besides the number written on the scale bar. Alternatively, mention it in the figure caption. In Figures 1 and 4, the abbreviations D0-D14 should be defined in the figure caption. Also, the acronyms used on the individual panels from Figs. 5 and 6 need to be explained in detail, both in the main text and figure caption. Also, it is not clear to me what is meant by "without colour" on line 312. 

We had included the amendments suggested. Referring to the description as “no colour” for tumoral cells, an explanation has been included both in the legend and in the text.

Additionally, the presence of tumoral cells within the core cannot be overlooked, particu-larly in the SK-HEP1 co-culture, where proliferative cells infiltrate and disrupt the integrity of the inner core. It is noteworthy that for TMCRE_SKHEP1 mixed spheroids, numerous tumoral cells appeared uncolored, as the initial labelling became diluted with each cell division, after 7 days of culture.

  1. The Discussion would further benefit from comparisons made with results obtained from different cancer models. For example, 3D bioprinted models have also been used to assess the response of cancer cells to chemotherapy [*5]. 

 [5*] Zhao Y, Yao R, Ouyang L, Ding H, Zhang T, Zhang K, et al. Three-dimensional printing of Hela cells for cervical tumor model in vitro. Biofabrication. 2014;6(3):035001.

Thank you once more for the valuable references and input. We have incorporated the results concerning HELA cells and expanded the discussion by including a paragraph dedicated to the limitations of the study. In this section, we also compare spheroids to alternative 3D tumor models.

“It should be highlighted that, in addition to spheroids, various approaches for complex cultures have been tested. In these cultures, not only primary hepatocytes but also tumor cells may exhibit enhanced functionality. For instance, increased metalloproteinase pro-duction was observed in bioprinted HELA cells compared to their 2D counterparts [67]. (…)

While tumor spheroids with enhanced functionality have been employed in studies of fundamental cancer biology [91] as well as drug screening and validation [92], they con-tinue to lack the major extracellular matrix component of the tumor microenvironment [3]. Alternative approaches include porous scaffolds, which have shown enhanced prolifera-tion, growth factor and cytokine secretion, and vascularization compared to both 2D monolayer cultures and 3D tissues embedded in Matrigel, both in vitro and after in vivo implantation [93]. Recently, the bioprinting of hepatocarcinoma models using HepG2 and human umbilical vein endothelial cells has emerged as a highly scalable platform for generating engineered tissue constructs containing vascular channels and multiple cell types [5].”

Reviewer 3 Report

Comments and Suggestions for Authors

This is a high-quality report on the usefulness of hepatocyte spheroids in assessing chemotherapy in hepatocellular carcinoma. The methodology is sophisticated. the quality of the presentation is high and written.

1. Please give information on what mixed spheroids were  prepared, provide the passage number of used cells
2. provide information on why this mouse strain was used
3. provide study limitations

Author Response

Review 3

This is a high-quality report on the usefulness of hepatocyte spheroids in assessing chemotherapy in hepatocellular carcinoma. The methodology is sophisticated. the quality of the presentation is high and written.

Thank you for the positive feedback and valuable advice. We have made efforts to enhance the draft based on the referee's suggestions

1. Please give information on what mixed spheroids were prepared, provide the passage number of used cells

Information has been provided in the M&M section

“Tumoral cell lines HepG2 (ATCC® No. HB-8065), SK-HEP1 (ATCC® No. HTB-52™), and HUH7 (No. JCRB0403) were cultured for maintenance in DMEM media supplement-ed with 10% FBS (ThermoFisher) and an antibiotic mixture (penicillin, streptomycin, am-photericin (ThermoFisher)) in a humidified incubator at 37°C with 5% CO2. To culture mixed spheroids, only tumoral cell at passages below 10 subcultures were employed”

  1. provide information on why this mouse strain was used

In The M&M, now the text reads:

“Primary hepatocytes were obtained from the strain resulting of the cross of Albu-min-Cre (B6.Cg-Speer6-ps1Tg(Alb-cre)21Mgn/J) and B6.129(Cg)-Gt(ROSA)26Sortm4(ACTB-tdTomato,-EGFP)Luo/J (or mT/mG), both purchased from the Jackson Laboratory. The mT/mG mouse serves as a double-fluorescent Cre reporter, expressing membrane-targeted tandem dimer Tomato (mT) before Cre-mediated excision and membrane-targeted green fluorescence protein (GFP) (mG) af-ter excision [28]. The B6J inbred mouse background is among the most frequently used strains due to its stability and ease of breeding. Many transgenic mice, including those generated using the Cre-lox system, such as the one employed in this study, are produced with a B6J background [29]. Upon crossing with an animal expressing the CRE protein under the Albumin promoter, we obtain an animal with GFP targeted to membranes in all cells expressing albumin, such as hepatocytes. This allows us to distinguish hepatocytes from other cells present in the cultures, such as tumoral cells, and also enables us to track hepatocyte-derived EVs in the culture media. Throughout the article, we will refer to this F1 as TMCRE.”

  1. provide study limitations

We have now included a paragraph of limitations, involving different topics:

“4.5 Limitations of the study

We acknowledge that the study had limitations concerning both the cell mixture and the analysis conducted to monitor chemotherapy outcomes. Regarding the cell mixture, many aspects of liver functionality were not replicated, such as the involvement of the immune system and the participation of stellate cells, which could be decisive in tumor progression [89]. The study would also benefit from incorporating a measurement of spheroid mass, which could provide insights into changes in cell density during growth and treatment. Indeed, a parameter of density could reflect modifications in the inner cel-lular composition and intercellular connection network [38]. However, such measure-ments require microfluidic devices, wherein single spheroids enter a designed analysis flow-channel dedicated to assessing their terminal velocity [90]. With the aid of these spe-cialized devices, values for weight, size, and mass density of the 3D biological sample could be accurately measured. In the same way, the metabolic capabilities of the spheroid and the impact of chemotherapy on both tumoral cells and hepatocytes should be further investigated in detail, including the specific mechanisms of cell death. While tumor spheroids with enhanced functionality have been employed in studies of fundamental cancer biology [91] as well as drug screening and validation [92], they continue to lack the major extracellular matrix component of the tumor microenvironment [3]. Alternative ap-proaches include porous scaffolds, which have shown enhanced proliferation, growth factor and cytokine secretion, and vascularization compared to both 2D monolayer cul-tures and 3D tissues embedded in Matrigel, both in vitro and after in vivo implantation [93]. Recently, the bioprinting of hepatocarcinoma models using HepG2 and human um-bilical vein endothelial cells has emerged as a highly scalable platform for generating en-gineered tissue constructs containing vascular channels and multiple cell types [5]. Addi-tionally, regarding EV release, it would be valuable to explore changes in the subpopula-tion of vesicles and the presence of markers of cell death following treatment, for both hepatocytes and tumoral cells. However, despite the limitations listed, the simplicity of the proposed model offers advantages in terms of implementation and result analysis.”

Reviewer 4 Report

Comments and Suggestions for Authors

The paper submitted could be a gentle contribution to whom is working with cancer sheroid model as more realistic subject of investigation for studying chemotherapy effectiveness.  The methods used and the laboratory work by the authors are very fine according to me. 

I think a major focus should be present for the different obtained mixed spheroids, because just this can be the novelty of the paper. 

Importantly, the authors proposed a scaffold-free model that is more relevant than materiel or agarose embedded spheroids fo HTS scalable testing. 

Many points should be addressed here and are requested to be revised,  in order to have a suitable paper to publish fo this referee.

I suggest to improve or correct the following points, as following:

1) abstract, it is inconsistent and not so appropriate. from "(2) METHODS; Mouse hepatocytes in primary cultures exhibit self-organization into spheroids when cultured in ultra-low attachment plates. (3) RESULTS; In this configuration, an ac- tive outer layer establishes a boundary with the media, while the inner core comprises a mass of cell debris." this will be changed and completely revised (not for the language).

A "ULA-using" related sentence cannot be a method in a abstract, sorry. In addition the vesicles aspect should be presented as "a result" point in the abstract if it is altro introduced in the introduction of the manuscript later (lines 67-71) .Thank you

2) Introduction, lines 27 -40 , please refine the English

3) you can already revise helping the editorial team, by checking all the superscript etc (eg. CaCl2)

4) I guess Cultex is Cultrex , line 115

5)p150 petri, what is p? adDMEM-F12 stands for "Advanced"? please take care of suffix and glossary not common for bioengineering and engineering investigators but likely known to biologists and biotechnologists

6) you can add ratio description, in M&M, "then 500 cells were added and mixed simultaneously with 5000 primary mouse hepatocytes" and when it will be relevant (maybe in Figures?). It is 1:10 ratio, tumoral vs primary cells, right? thank you, if you so remind the reader the content of mixed spheroid like its ratio could be important for personal scientific consideration of the results. 

7) HamsterAr wjat is "Ar"? 

8) antibodies called "anti XX" should be written as "anti-XX"

9) figure 3 presentation of the per data are correct, but not so intuitive to me. Why cannot you set T0 value equal to 1 instead of 0? then you can clearly see the 2D levels vs 3D levels of the relative quantification of the target gene? 

10) you can better allow the reader to distinguish between all spheroids analyzed by calling them mono-culture spheroids or co-cultured (mixed) spheroids. You should stress about the importance and the challenge to for and monitor different aggregate of different cell types and primary cells ( as already seen in other spheroids made for regenerative medicine purposes and using this terminology. please check as reference and useful source for discussion Paris et al 2023, published in the same journal at doi.org/10.3390/bioengineering10020189. in this paper also the organization of the co-cultured cell type results in surface vs core for the 2 cell types cocultured, as the authors here similarly report for their mixed spheroid (lines 310 -312). 

11) figure 5 should be completely revised. the four images In the figures needs panel numbering or letter indications, to be then described in the figure 5 legend. 

12) figure 5 legend, the first picture is monoculture of TMCRE (primary hepatocytes control I guess) and not one mixed spheroid..., so you should change the description that is currently incorrect. please, also add title of figure 5 , then its description, as you've done for other figures.

13) Figure 5 miss length micron bar in all 4 images (panels). please add them

14) about Figure 6 legend,  authors wrote "The growth curve was constructed by measuring spheroids over time, highlighting variations in their growth rates compared to hepatocyte spheroids." I believe it will be better if authors mention that it is the diameter actually measured. or not? also correct the y-axis in the plot as "mean diameter length (micron), if it is. 

15) For Figure 7, if I am not wrong, in Fig 7 legend you can remember that hepatocytes are green because are GFP-cells from in vivo GMO mice.also because in the results there is Sytox mention, which you can understand is not green but blue only if you go back to M&M section (or remembering/reading Fig2 legend).

16) figure 8 resolution is poor in the pdf I read, sorry. can you improve it submitting a higher quality image?

17) what about CD81 staining for vesicles? can you say something or providing other supplementary info? 

18) about Figure 4, what are D1, D4, D7...etc? days of culture? dilutions? please add in the legend and refer the meaning also at line 299-300

19) I don't find the method used to disaggregate mixed spheroids (trypsinization or other enzymes? time? agitation? temperature?) before the flow cytometer analysis and gating strategy to assess the differences (drugs and % of figure 8) . can you please write or please tell me where it is? thanks you

20) the rational for the used dose of the chemiotherapic drug should be expressed. this is important also because the high toxicity found in normal hepatocytes for Sorafenib in your mixed model 

21) In addition to what authors already tell us in the discussion (from line 430) ì, please add more relevant information about the tumor cell lines used in your study. Which are their histology and cytology differences among them (cell junctions, proliferative rate, polarity, antibody markers, noted chemo-resistance in 2D cultures by literature.)?

22) honestly, we saw live and dead cells from Sytox, but a metabolic based assay, like MTT or Alamarblue or ATP-measuring assay would be enhance the significance and should be correlated to the drug sensitivity results found by the authors. Can you perform this if major revision is requested? alternatively a hepatocyte functionality test would be also a benefit (specific secretion of primary hepatocyte protein or molecules)

23) did you have data about metastatic marker expression challenge after mixes co-cultured spheroids drug treatment? 

24) in the discussion, if authors agree, they can consider to propose about mass density of the drug-treated spheroids as a valuable indicator of spheroid state that could be useful in following up samples and their change in a future reproducible manner; I recommend this indeed, because mass density changes according to diameter, cell viability and intrinsic physical properties of the single cells, including the ones in the core of spheroids. Please have a look at Marrazzo et al. 2021 for replying and discuss in the manuscript. ( doi 10.3390/antibiotics10070750, this will be a suggested reference).

Comments on the Quality of English Language

only some parts, like in the introduction and abstract should be upgraded. 

Author Response

Review 4

The paper submitted could be a gentle contribution to whom is working with cancer sheroid model as more realistic subject of investigation for studying chemotherapy effectiveness.  The methods used and the laboratory work by the authors are very fine according to me. 

I think a major focus should be present for the different obtained mixed spheroids, because just this can be the novelty of the paper. 

Importantly, the authors proposed a scaffold-free model that is more relevant than materiel or agarose embedded spheroids fo HTS scalable testing. 

Many points should be addressed here and are requested to be revised, in order to have a suitable paper to publish fo this referee.

We appreciate the referee's comments and valuable suggestions. We've tried to address all raised questions, and we hope the paper could be accepted for publication.

I suggest to improve or correct the following points, as following:

1) abstract, it is inconsistent and not so appropriate. from "(2) METHODS; Mouse hepatocytes in primary cultures exhibit self-organization into spheroids when cultured in ultra-low attachment plates. (3) RESULTS; In this configuration, an ac- tive outer layer establishes a boundary with the media, while the inner core comprises a mass of cell debris." this will be changed and completely revised (not for the language).

A "ULA-using" related sentence cannot be a method in a abstract, sorry. In addition the vesicles aspect should be presented as "a result" point in the abstract if it is altro introduced in the introduction of the manuscript later (lines 67-71) .Thank you

Following the referee´s advice, we have rewritten the abstract to provide greater consistency in its structure. Additionally, we have worked to separate the method and results sections to enhance the clarity of the text. Final text read as follow:

“Abstract: (1) BACKGROUND: Three-dimensional cellular models provide a more comprehensive representation of in vivo cell properties, encompassing physiological characteristics and drug susceptibility. (2) METHODS: Primary hepatocytes were seeded in ultra-low attachment plates to form spheroids, with or without tumoral cells. Spheroid structure, cell proliferation, and apoptosis were analyzed using histological staining techniques. In addition, extracellular vesicles were isolated from conditioned media by differential ultracentrifugation. Spheroids were exposed to cytotoxic drugs, and both spheroid growth and cell death were measured by microscopic imaging and flow cytometry with vital staining, respectively. (3) RESULTS: Concerning spheroid structure, an active outer layer forms a boundary with the media, while the inner core comprises a mass of cell debris. Hepatocyte-formed spheroids release vesicles into the extracellular media, and a decrease in the concentration of vesicles in the culture media can be observed over time. When co-cultured with tumoral cells, a distinct distribution pattern emerges over the primary hepatocytes, resulting in different spheroid conformations. Tumoral cell growth was compromised upon anti-tumoral drug challenges. (4) CONCLUSION: Treatment of mixed spheroids with different cytotoxic drugs enables the characterization of drug effects on both hepatocytes and tumoral cells, determining drug specificity effects on these cell types.”

2) Introduction, lines 27 -40 , please refine the English

The paragraph has been rewritten, and now reads as follows:

“For some time now, the recognition that 2D monolayer cells on plastic plates do not fully represent various aspects of the in vivo environment has prompt the development of better hepatocyte models for drug screening [1]. Recognizing the critical role of the cellular environment in shaping cell behavior, significant efforts have been made to recreate the stem cell niche in laboratory settings, aiming for a more precise physiological reproduction of spatiotemporal cell–cell interactions, understanding this concept as the evolving relationships between various cell types within the structure, including their spatial organization, as the culture evolve [2]. Pursuing this objective, there is a growing trend to-wards recreating in vitro as much of the tissue architecture and function observed in vivo as possible. However, demonstrating the physiological relevance of increasingly complex models presents considerable challenges. For tumoral models, one promising approach is to recapitulate the microenvironment surrounding tumors, including vascularization, stromal cells, immune cells, cancer-associated fibroblasts, and microvascular cells, in addition to cancer epithelial cells [3]. Recreation of vascularization has been achieved in tumoral models by combining a bioprinting strategy of human umbilical vein endothelial cells with multicellular spheroids derived from glioma cells [4]. Another aspect is the compositional material of the microenvironment, which can be achieved with multimaterial bioprinting [5]. While bioengineers may aim to develop in vitro models that replicate specific tissue features relevant to physiological or diseased functions, a pragmatic perspective also supports simpler models for the majority of users. In this context, models featuring one or two types of cells in 3D culture models are often more robust for mechanistic studies and applications than their more intricate counterparts [6].”

3) you can already revise helping the editorial team, by checking all the superscript etc (eg. CaCl2)

Corrected

4) I guess Cultex is Cultrex , line 115

Corrected

5) p150 petri, what is p? adDMEM-F12 stands for "Advanced"? please take care of suffix and glossary not common for bioengineering and engineering investigators but likely known to biologists and biotechnologists

The nomenclature has been corrected. We would like to thank the referee for their careful review of the section, and we apologize for these oversights.

6) you can add ratio description, in M&M, "then 500 cells were added and mixed simultaneously with 5000 primary mouse hepatocytes" and when it will be relevant (maybe in Figures?). It is 1:10 ratio, tumoral vs primary cells, right? thank you, if you so remind the reader the content of mixed spheroid like its ratio could be important for personal scientific consideration of the results. 

The ratio has been explicitly stated in the Materials and Methods section and at the beginning of the Results section.

7) HamsterAr wjat is "Ar"? 

Now we corrected the nomenclature as “Armenian Hamster anti-Cd81”

8) antibodies called "anti XX" should be written as "anti-XX"

Thanks for the correction, we did it.

9) figure 3 presentation of the per data are correct, but not so intuitive to me. Why cannot you set T0 value equal to 1 instead of 0? then you can clearly see the 2D levels vs 3D levels of the relative quantification of the target gene? 

The graph displays differences in CT values rather than proportions or fold changes. The value 0 represents the CTs of mature hepatocytes when they are just disaggregated, relative to the reference gene Rplp0. Negative values indicate genes with decreased expression in culture, while positive values indicate increased expression. This way of presenting the data also allows for a similar scale for comparing 2D and 3D cultures, regardless of whether gene expression increases or decreases. If T0 is set to 1, as taking the exponent, then genes with increased expression would range from 1 to infinity, while those with decreased expression would range between 0 and 1. Certainly, we would prefer to maintain this data representation. However, we have added a sentence in the text to make this presentation clearer.

“Transcription analysis revealed that after 7 days of culture, there is a reduction in the expression of liver markers compared to primary hepatocytes immediately after liver per-fusion (T0). In Figure 3, we present the ddCt values, where zero represents the expression level of T0 relative to the reference gene. Therefore, negative values indicate a decrease in the expression of those genes over time in culture, while positive values are obtained for genes whose expression increase in vitro culture. It can be observed that the expression in spheroids (3D) is more similar to that of freshly isolated hepatocytes compared to 2D cultures for the same duration. However, while the expression of some markers such as Alb or Hnf4 is preserved in spheroids, there is a decrease in enzymes associated with hepatocyte energy metabolism (G6pc and Gck), as well as a notable drop in Cyp3a11. Nevertheless, Cyp3a11 expression remains more abundant in spheroids compared to hepatocytes cultured in 2D.”

10) you can better allow the reader to distinguish between all spheroids analyzed by calling them mono-culture spheroids or co-cultured (mixed) spheroids. You should stress about the importance and the challenge to for and monitor different aggregate of different cell types and primary cells ( as already seen in other spheroids made for regenerative medicine purposes and using this terminology. please check as reference and useful source for discussion Paris et al 2023, published in the same journal at doi.org/10.3390/bioengineering10020189. in this paper also the organization of the co-cultured cell type results in surface vs core for the 2 cell types cocultured, as the authors here similarly report for their mixed spheroid (lines 310 -312). 

Thank you very much for the suggestions. Although we have mentioned the term of co-culture, now we emphasize the concept of mono-culture along the text, and we had referenced the article by Paris et al, respect to the multicellular organization in the spheroid.

“In respect to co-culture spheroid, our observations revealed a distinct peripheral dis-tribution of tumoral cells, leaving primary hepatocytes forming an inner core. This spe-cialized distribution of co-cultured cells has been observed previously, where cells dis-playing epithelial cell markers accommodate themselves in the periphery of the spheroid [68].”

11) figure 5 should be completely revised. the four images In the figures needs panel numbering or letter indications, to be then described in the figure 5 legend. 

Done

12) figure 5 legend, the first picture is monoculture of TMCRE (primary hepatocytes control I guess) and not one mixed spheroid..., so you should change the description that is currently incorrect. please, also add title of figure 5 , then its description, as you've done for other figures.

The legend has been rewritten:

“Figure 5. Tridimensional reconstruction of a series of confocal images taken from intact spheroids cultured for 7 days. Panel A shows the reconstruction of a mono-culture spheroid formed by primary hepatocytes (TMCRE), while Panels B, C, and D depict mixed spheroids containing tumoral cells HEPG2 (TMCRE_HEPG2), SKHEP-1 (TMCRE_SKHEP1), and HUH7 (TMCRE_HUH7), respectively. Tumoral cells, labelled before spheroid formation, may appear either reddish or uncoloured (see text for details), while hepatocytes exhibit green fluorescence.”

13) Figure 5 miss length micron bar in all 4 images (panels). please add them

Added

14) about Figure 6 legend,  authors wrote "The growth curve was constructed by measuring spheroids over time, highlighting variations in their growth rates compared to hepatocyte spheroids." I believe it will be better if authors mention that it is the diameter actually measured. or not? also correct the y-axis in the plot as "mean diameter length (micron), if it is. 

Thanks for the comment, we include the word diameter in the legend and in the figure.

15) For Figure 7, if I am not wrong, in Fig 7 legend you can remember that hepatocytes are green because are GFP-cells from in vivo GMO mice.also because in the results there is Sytox mention, which you can understand is not green but blue only if you go back to M&M section (or remembering/reading Fig2 legend).

We have now included a reminder in the legends of Figures 2 and 7 stating that hepatocytes obtained from TMCRE mice are labeled green because they express GFP bound to a membrane peptide. Similarly, we mention in the text that Sytox is a cell death stain that fluoresces at 450 nm

16) figure 8 resolution is poor in the pdf I read, sorry. can you improve it submitting a higher quality image?

We solve the problem, thanks for pointing out.

17) what about CD81 staining for vesicles? can you say something or providing other supplementary info? 

We have confirmed the presence of CD81 in EV preparations obtained from spheroid conditioned media, as shown in Supplemental Figure 1. We did not perform CD81 staining of EVs for flow cytometry, as labelling EVs with antibodies is more likely to generate artifacts and false positives.

18) about Figure 4, what are D1, D4, D7...etc? days of culture? dilutions? please add in the legend and refer the meaning also at line 299-300

We have corrected the label and added an explanation to the text. The legend now reads as follows:

“Figure 4. Fluorescence events concentration in the culture media obtained from spheroids along the culture time (n= 3). The cell media was sampled at days 1(D1), 4 (D4), 7(D7), 10 (D10), 14(D14) and 17(D17) after cell seeding.”

19) I don't find the method used to disaggregate mixed spheroids (trypsinization or other enzymes? time? agitation? temperature?) before the flow cytometer analysis and gating strategy to assess the differences (drugs and % of figure 8) . can you please write or please tell me where it is? thanks you

The method for disaggregation can be found in the M&M section:

“2.4 Flow cytometry

After a 20-minute incubation in Tryple (Gibco, ThermoFisher Scientific) at cell incubator, spheroids were mechanically disaggregated with a 1000 µL pipette. Cells were washed with eBio-science™ Flow Cytometry Staining Buffer and suspended in a fluid containing a working concentration of SYTOX® Blue Dead Cell Stain (ThermoFisher Scientific).”

20) the rational for the used dose of the chemiotherapic drug should be expressed. this is important also because the high toxicity found in normal hepatocytes for Sorafenib in your mixed model 

We assayed a high dose, which has been observed to be effective in previous studies, as mentioned in the Materials and Methods section. Our aim was to use a high dose to evaluate the model's effectiveness in determining cell toxicities. We have included this idea in the discussion.

“According to previous studies, the dose we have employed is relatively high [55] [32] [33], which might also explain the observed effects on hepatocytes. The rationale behind this choice was to achieve a high clearance of tumoral cells from spheroids, in order to obtain clear results regarding the utility of mixed spheroids as a tool.”

21) In addition to what authors already tell us in the discussion (from line 430) ì, please add more relevant information about the tumor cell lines used in your study. Which are their histology and cytology differences among them (cell junctions, proliferative rate, polarity, antibody markers, noted chemo-resistance in 2D cultures by literature.)?

Thank you for the suggestion. We have expanded the characterization of the tumoral cell lines discussed, which become a section in the discussion. The provided references can assist readers in exploring additional details elsewhere.

4.3 Tumoral cells assayed

In respect to co-culture spheroid, our observations revealed a distinct peripheral distribution of tumoral cells, leaving primary hepatocytes forming an inner core. This specialized distribution of co-cultured cells has been observed previously, where cells dis-playing epithelial cell markers accommodate themselves in the periphery of the spheroid [68]. The mechanism behind cell distribution requires further analysis, but we have also observed differences between tumoral cell lines. The SK-HEP1 cell line, suspected to derive from an endothelial origin [69], exhibits the fastest growth rate and completely covers the surface of the spheroid. This cell line has metastatic potential, as it has been reported to engraft in nude mice [70]. Additionally, it demonstrates higher resistance to methotrexate compared to other employed cell lines [71]. Interestingly, SK-HEP1 synthesizes various proteins regulating cell attachment, including vimentin [69] and fibronectin [72], with the latter production enhanced by extracellular matrix components [73]. However, it has been described that SK-HEP1 does not exhibit polarization [74], while HUH7 has shown polarity in Matrigel cultures [75], and HepG2 has been described to have a polarization some-what similar to primary hepatocytes [76]. For the latter, they show typical cell junction proteins such as claudin, e-cadherin, or occluding [77]. However, HepG2 and HUH7 exhibit differences in their integration levels with primary hepatocytes, which are challenging to explain based on their inherent characteristics. Both cell lines exhibit variations in the expression of several protein profiles compared to mature hepatocytes [78], yet they share high similarities in their metabolic characteristics [79]. While their sensitivity to sorafenib in 3D spheroids has been previously described, showcasing similarities in the re-duction of cell growth in the presence of the chemotherapy drug [80], their response to collagen presence or environmental stiffness remains consistent between the two [81, 82]. HepG2 is also able to form tumors in nude mice at a higher rate than SKHEP1, and it shows less drug inhibition by different chemotherapeutic agents [70]. Notably, metastatic invasiveness could be increased by enriching the population of cancer stem cells through spheroid formation [83]. It is also important to consider that HepG2 presents expression related to fetal development, such as high expression of insulin-like growth factor-1 (IGF-1), which leads to the overactivation of molecular targets associated with the insulin pathway [84]. On the contrary, HUH7 does not exhibit these characteristics regarding insulin pathway activity; however, it has demonstrated a high correlation in gene expression spectrum with drug-metabolizing enzymes, transporters, and glutathione-S-transferase activity compared to primary human hepatocytes [85]. It is interesting to highlight that all three cell lines show typical HCC markers such as AFP, EpCAM, or Cd133 [86][87]. Speculatively, we attribute the differences in their aggregation with primary hepatocytes to their distinct origins, with HepG2 being a hepatoblastoma-derived cell line and HUH7 derived from hepatocarcinoma cell lines, showing disparities in certain receptors and responses to fibroblast growth factors 1 or 2 [88].”

22) honestly, we saw live and dead cells from Sytox, but a metabolic based assay, like MTT or Alamarblue or ATP-measuring assay would be enhance the significance and should be correlated to the drug sensitivity results found by the authors. Can you perform this if major revision is requested? alternatively a hepatocyte functionality test would be also a benefit (specific secretion of primary hepatocyte protein or molecules)

Thank you very much for these important questions and suggestions. There are several published protocols for performing MTT assays on spheroids; however, they are technically challenging, and so far, we have not obtained consistent results. On the other hand, we have attempted the LDH assay, but due to the inherent cell death in hepatocyte spheroids, there are elevated values of LDH even in untreated spheroids. Additionally, these tests do not allow us to distinguish between damaged tumoral cells and hepatocytes. While we certainly aim to make progress in this direction, at present, we have not been able to identify a satisfactory complementary assay. Nonetheless, the main assertion—that cells are undergoing damage after treatment and that the effect on tumoral and primary hepatocytes can be differentiated—is well-covered by the proposed assay, so we have retained this assay in the revised version.

23) did you have data about metastatic marker expression challenge after mixes co-cultured spheroids drug treatment? 

This is indeed a valuable suggestion, and it would be worthwhile to explore under more complex culture conditions, such as Matrigel cultures. However, we do have concerns that the concept of metastasis in the context of spheroid growth may be viewed as pushing the limits of the model. Nevertheless, we will keep it in mind for future assays.

24) in the discussion, if authors agree, they can consider to propose about mass density of the drug-treated spheroids as a valuable indicator of spheroid state that could be useful in following up samples and their change in a future reproducible manner; I recommend this indeed, because mass density changes according to diameter, cell viability and intrinsic physical properties of the single cells, including the ones in the core of spheroids. Please have a look at Marrazzo et al. 2021 for replying and discuss in the manuscript. ( doi 10.3390/antibiotics10070750, this will be a suggested reference).

We thank the referee for introducing us to this important concept. We have included it in the discussion and also mentioned it in the section discussing the limitations of the study.

“Although not considered in our study, the degree of compactness of the spheroids will also be an important measure of spheroid activity since they reveal the collective variations in the single cells comprised in them, which are connected to growth and cell cycle changes [38].”

“Regarding the cell mixture, many aspects of liver functionality were not replicated, such as the involvement of the immune system and the participation of stellate cells, which could be decisive in tumor progression [89]. The study would also benefit from incorporating a measurement of spheroid mass, which could provide insights into changes in cell density during growth and treatment. Indeed, a parameter of density could reflect modifications in the inner cellular composition and intercellular connection network [38]. However, such measurements require microfluidic devices, wherein single spheroids enter a designed analysis flow-channel dedicated to assessing their terminal velocity [90]. With the aid of these specialized devices, values for weight, size, and mass density of the 3D biological sample could be accurately measured.”

Reviewer 5 Report

Comments and Suggestions for Authors

The manuscript entitled " Three-Dimensional Hepatocyte Spheroids: Model for Assessing Chemotherapy in Hepatocellular Carcinoma " is focused on an interesting topic with numerous implications both in the research and practical fields.

In general, the work needs to be reviewed from different points of view.

In the current format the paper cannot be published.

The authors provide a lot of information and, in some cases, the discussion is not very clear and linear.

First of all, what is the aim of the study? It is not clear.

Introduction

Line 32           What do the authors mean with the word “spatio-temporal”?

Line 47 - 49    This sentence is unclear. What does it mean “for mice”?

Line 49-51      This concept needs to be explained, as it is not clear in this context. Authors talk about the structure of the spheroid without having introduced in the previous sentence a minimum of how it is formed, etc.

Line 64            It is necessary to better define the aim of the study and what the authors expect to gain from the work.

Line 66            What does it mean “automatable”?

Materials and Methods

It is necessary to describe the analysis times in detail, as it is not clear how long the spheroids were kept in culture and the times at which the observations were carried out.

The procedures applied for treatment with anti-tumor active ingredients are completely lacking. Characteristics, dosages, choice of anticancer drugs, etc.

Line 112          What was the percentage of vitality? Has it been calculated? How long were they kept in culture? Results are shown at different times at which the samples were stained. You also need to include it in M&M.

Line 119          What was the procedure to change the medium? Is it possible to have a cell loss during this procedure? Have the authors verified this possibility?

Line 122          The features of the three cell lines should be entered into a table, in order to understand if there are any differences between them.

Line 128-134  It is necessary to detail the features of the drugs used and what effect they can have on tumour cells; it would be better to write a dedicated chapter or a table.

Results

Linea 248       Indicate in the figure with arrows where the necrosis can be seen.

Line 250          The described observations are not as evident in the figure. This should be highlighted more.

Line 270         Where in M&M do the authors talk about 2D cultivation?

Line 286         Why were these markers considered? What do they highlight?

Line 310-312  This statement needs to be supported by more data and considerations.

Line 338-348  It is not specified how many replications were evaluated. How were the results processed? Where are described in details the chemical compounds?Discussion and Conclusions

The conclusion chapter is poorly organized: the purpose of the work is not specified, the conclusions drawn by the authors are not sufficiently supported by the results of the work. Furthermore, not all results are discussed and compared with each other.

It would be useful to write an introductory sentence to contextualize the study.

An extensive review is necessary. It would be appropriate to rewrite the chapter

Figures

Figure 5         The caption needs to be more detailed

Author Response

Review 5

The manuscript entitled " Three-Dimensional Hepatocyte Spheroids: Model for Assessing Chemotherapy in Hepatocellular Carcinoma " is focused on an interesting topic with numerous implications both in the research and practical fields.

In general, the work needs to be reviewed from different points of view.

In the current format the paper cannot be published.

The authors provide a lot of information and, in some cases, the discussion is not very clear and linear.

First of all, what is the aim of the study? It is not clear.

We appreciate the reviewer's effort in reading and identifying the weakest points of our work. We have attempted to clarify the aspects they highlighted, and together with responses to other reviewers' comments, both the introduction and discussion sections have been extensively rewritten. The objective of the study was to characterize a 3D culture model comprising primary hepatocytes and HCC cell lines, which closely mimics physiological conditions compared to 2D cultures. This model can be utilized to investigate various aspects of tumoral cell lines, including their responses to chemotherapy treatments, in a more specific manner than mono-cultures or 2D cultures. We hope the reviewer will find the revised version suitable for publication.

Introduction

Line 32           What do the authors mean with the word “spatio-temporal”?

Spatio-temporal refers to the relationships and interactions between different types of cells within an organized structure, which evolve as cells grow or are treated. For instance, in the formation of spheroids, it involves the specialization of different cell layers based on their positions and interactions with other cells. To state in a more clear matter, we have now rephrase the sentence as follows:

“Recognizing the critical role of the cellular environment in shaping cell behavior, significant efforts have been made to recreate the stem cell niche in laboratory settings, aiming for a more precise physiological reproduction of spatio-temporal cell–cell interactions, understanding this concept as the evolving relationships between various cell types within the structure, including their spatial organization, as the culture evolve [2].”

Line 47 - 49    This sentence is unclear. What does it mean “for mice”?

Arguably the most important cytochromes in the human liver is CYP3A4. Its counterpart in mice is CYP3A11. We've rephrased the sentence for clarity.

Line 49-51      This concept needs to be explained, as it is not clear in this context. Authors talk about the structure of the spheroid without having introduced in the previous sentence a minimum of how it is formed, etc.

Now the paragraph had some references about the formation of spheroids, and the differences between spheroids and cell aggregates.

One of the models that combine effectiveness with simplicity is spheroid cell culture [7]. In the absence of surface adhesion, spheroid formation is a forced phenomenon de-pendent on adhesion molecules like integrins [14]. Cell spheroids exhibit a structure with elongated cells on the surface and layers of cells inside, comprising both growing and non-growing cells, as well as cells experiencing low oxygen levels [15]. An important concept is to distinguish between cell aggregates and spheroids, as the latter involve specific cell-cell interactions within a structured 3D framework and consistent geometry, leading to the development of pathophysiological gradients within the spheroid, which are largely influenced by their diameter [8]. Spheroids smaller than 150 µm display 3D cell-cell and cell-matrix interactions, yet distinct radial proliferative and pathophysiological gradients may not yet be fully evident, typically observed in spheroids ranging from 200 to 500 µm [8]. The 3D spheroid culture of primary hepatocytes, without the addition of extracellular matrices or scaffolds, has been characterized for various pharmacological applications [9] [10]. When compared to 2D monolayer or sandwich-cultured primary hepatocytes, these spheroids maintain a more physiologic phenotype of native hepatic tissue at both protein and RNA levels, as evidenced in hepatotoxicity assays [11].”

Line 64            It is necessary to better define the aim of the study and what the authors expect to gain from the work.

We had reformulated the aims of the work, in order to be more precise:

“In the current study, the main goal is to generate reproducible spheroids, as a 3D culture model potentially suitable for automation. This model allows to maintain primary hepatocytes closer to their differentiated state and for co-culture with hepatocarcinoma cells. Through this approach, the specific effects of anti-tumoral treatments could be investigated in a more physiological manner compared to 2D mono-cultures. Additionally, it allows for the simultaneous study of the effects on both primary and tumoral cells.”

Line 66            What does it mean “automatable”?

In this context, 'automatable' refers to the possibility of carrying out certain tasks with minimal human intervention using mechanical systems or machinery. We had rephrased the sentence to clearly state that what is automatable is the culture process itself, not the spheroid.

Materials and Methods

It is necessary to describe the analysis times in detail, as it is not clear how long the spheroids were kept in culture and the times at which the observations were carried out.

We had included the following sentence:

“The duration of the cultures was variable according to the experiments, ranging from 7 to 21 days, and it is described for each study in the text of results section and it in the figure legend.”

The procedures applied for treatment with anti-tumor active ingredients are completely lacking. Characteristics, dosages, choice of anticancer drugs, etc.

The procedure and dosage are described in the M&M section:

“For cytotoxic drug treatment, the compound was added to the spheroid growth media af-ter day four of spheroid formation and refreshed at the same time as the control. The drugs employed and their concentrations were sorafenib (Raybiotech) at 50 µM [31], dacarbazine at 50 µM (Raybiotech) [32], and methotrexate (Sigma) at 50 µM [33].”

Addition information about drug dosage election is included in the discussion:

“According to previous studies, the dose we have employed is relatively high [55] [32] [33], which might also explain the observed effects on hepatocytes. The rationale behind this choice was to achieve a high clearance of tumoral cells from spheroids, in order to obtain clear results regarding the utility of mixed spheroids as a tool.”

Line 112          What was the percentage of vitality? Has it been calculated? How long were they kept in culture? Results are shown at different times at which the samples were stained. You also need to include it in M&M.

The number of cells refers to vital cells, determined by simple trypan blue staining at the time of cell seeding. The percentage of vital cells in primary cultures varies widely among preparations, but we have worked with preparations having over 70% vitality. Additionally, statements indicating cell vitality have been included, along with the note about the duration of cultures mentioned earlier:

“Only those preparations with vitality over 70%, according to the staining with Trypan Blue Solution 0.4% (ThermoFisher Scientific) were employed for subsequent procedures.”

“Spheroids were formed in Nunclon Sphera 3D Plates with Low Attachment and U-Bottom 96-well Plates (Nunc, ThermoFisher Scientific) by seeding 5000 vital hepato-cytes per well (negative for staining with Trypan Blue Solution 0.4%) in a 200 µL volume of a specific media…”

Line 119          What was the procedure to change the medium? Is it possible to have a cell loss during this procedure? Have the authors verified this possibility?

The media was replaced 40% each time to minimize disturbance to the cells. The first replacement occurs on day 3, once spheroids are formed, ensuring minimal disruption as the spheroids remain intact after media addition. Certainly, any cells unable to integrate into the spheroid structure will be subsequently eliminated. As the spheroids exhibited uniform size, we can infer that a similar number of cells integrated into the spheroid across all wells. We added the information in the M&M.

Line 122          The features of the three cell lines should be entered into a table, in order to understand if there are any differences between them.

Thank you for the suggestion. One challenge we face is the abundance of information already present in our manuscript - comprising 8 figures and 5 supplementary figures. The table you requested would be more suitable for a review article focused on cancer cell lines, which are not the primary focus of our paper. Our emphasis lies in constructing a model applicable to HCC cell lines, for which we have selected three well-established models. Nonetheless, we have incorporated references and insights into the discussion, dedicating a separate section, that underscore the distinctions among these cell lines.

4.3 Tumoral cells assayed

In respect to co-culture spheroid, our observations revealed a distinct peripheral distribution of tumoral cells, leaving primary hepatocytes forming an inner core. This specialized distribution of co-cultured cells has been observed previously, where cells displaying epithelial cell markers accommodate themselves in the periphery of the spheroid [68]. The mechanism behind cell distribution requires further analysis, but we have also observed differences between tumoral cell lines. The SK-HEP1 cell line, suspected to derive from an endothelial origin [69], exhibits the fastest growth rate and completely covers the surface of the spheroid. This cell line has metastatic potential, as it has been reported to engraft in nude mice [70]. Additionally, it demonstrates higher resistance to methotrexate compared to other employed cell lines [71]. Interestingly, SK-HEP1 synthesizes various proteins regulating cell attachment, including vimentin [69] and fibronectin [72], with the latter production enhanced by extracellular matrix components [73]. However, it has been described that SK-HEP1 does not exhibit polarization [74], while HUH7 has shown polarity in Matrigel cultures [75], and HepG2 has been described to have a polarization somewhat similar to primary hepatocytes [76]. For the latter, they show typical cell junction proteins such as claudin, e-cadherin, or occluding [77]. However, HepG2 and HUH7 exhibit differences in their integration levels with primary hepatocytes, which are challenging to explain based on their inherent characteristics. Both cell lines exhibit variations in the expression of several protein profiles compared to mature hepatocytes [78], yet they share high similarities in their metabolic characteristics [79]. While their sensitivity to sorafenib in 3D spheroids has been previously described, showcasing similarities in the reduction of cell growth in the presence of the chemotherapy drug [80], their response to collagen presence or environmental stiffness remains consistent between the two [81, 82]. HepG2 is also able to form tumors in nude mice at a higher rate than SKHEP1, and it shows less drug inhibition by different chemotherapeutic agents [70]. Notably, metastatic invasiveness could be increased by enriching the population of cancer stem cells through spheroid formation [83]. It is also important to consider that HepG2 presents expression related to fetal development, such as high expression of insulin-like growth factor-1 (IGF-1), which leads to the overactivation of molecular targets associated with the insulin pathway [84]. On the contrary, HUH7 does not exhibit these characteristics regarding insulin pathway activity; however, it has demonstrated a high correlation in gene expression spectrum with drug-metabolizing enzymes, transporters, and glutathione-S-transferase activity compared to primary human hepatocytes [85]. It is interesting to highlight that all three cell lines show typical HCC markers such as AFP, EpCAM, or Cd133 [86][87]. Speculatively, we attribute the differences in their aggregation with primary hepatocytes to their distinct origins, with HepG2 being a hepatoblastoma-derived cell line and HUH7 derived from hepatocarcinoma cell lines, showing disparities in certain receptors and responses to fibroblast growth factors 1 or 2 [88].

Line 128-134  It is necessary to detail the features of the drugs used and what effect they can have on tumour cells; it would be better to write a dedicated chapter or a table.

For similar reasons as those outlined above, we have included information on the various drugs as part of the discussion, in a dedicated section:

“4.4 Chemotherapeutic agents employed

According to previous studies, the dose we have employed is relatively high [54] [31] [32], which might also explain the observed effects on hepatocytes. The rationale behind this choice was to achieve a high clearance of tumoral cells from spheroids, in order to obtain clear results regarding the utility of mixed spheroids as a tool. Regarding the drugs employed, sorafenib is a chemotherapy drug used against hepatocarcinoma [55]. Its mechanism of action relies on competitive inhibition of ATP binding to the catalytic domains of various kinases [56], subsequently blocking different signalling pathways such as PDGB (platelet-derived growth factor), vascular endothelial growth factor (VEGF),or Raf signalling [57]. On the other hand, methotrexate interrupts one-carbon metabolism by inhibiting the synthesis of tetrahydrofolate, thereby inhibiting the synthesis of purines and thymidines, impairing cell cycle progression. Cells become resistant by overcoming this limitation, for instance, by overexpressing enzymes of the one-carbon metabolism [58]. Finally, dacarbazine, an alkylating agent commonly used in combination with other chemotherapeutic agents, may act as a purine analogue and antimetabolite. Additionally, this drug is extensively metabolized in the liver and produces intermediates, some of which have alkylating activity, causing methylation, modification, and cross-linking of DNA, thereby inhibiting DNA, RNA, and protein synthesis [59]. Contrary to the activation of the drug observed in the liver for dacarbazine, the activity of tyrosine kinase inhibitors like sorafenib has been observed to decrease when CYP3A4 is active [60]. It should be noted that all three drugs can cause liver damage, and the effect over cells is assessable through various analytical approaches, ranging from high-throughput analysis, such as measuring size through automated high-content screening, to flow cytometry analysis for different cell populations. Flow cytometry provides insights into the damage affecting the inner core of hepatocytes induced by cytotoxic drugs. Regarding the mechanism of toxicity, it has been described for all three drugs. Dacarbazine can be metabolically toxic upon its transformation by CYP1A2 [52] [61], with the main contribution being the generation of reactive oxygen species [52]. Methotrexate is also converted in the liver, and its derivatives accumulate in different tissues, including the liver, causing damage upon exposure, which is also related to oxidative stress [62] [63]. Finally, sorafenib, like other tyrosine kinase inhibitors, has been observed to increase the risk of liver toxicity [64]. The main concern regarding their toxicity is once again the production of reactive metabolites leading to hepatocellular damage through oxidative stress, mitochondrial dysfunction, and impaired glycolysis [65]. As mentioned earlier, the metabolism of sorafenib is primarily catalyzed by CYP3A4 [60]. These dependence of the hepatocyte machinery justifies the application of 3D models, which exhibit a higher capacity for inducible expression of CYP3A4 compared to 2D cultures [10]. It should be highlighted that, in addition to spheroids, various approaches for complex cultures have been tested. In these cultures, not only primary hepatocytes but also tumor cells may exhibit enhanced functionality. For instance, increased metalloproteinase production was observed in bioprinted HELA cells compared to their 2D counterparts [66]. This underscores the rationale for utilizing 3D methods also to evaluate tumor cell behavior. Future studies, involving a larger number of drugs, could also benefit from this approach to refine the drug dosages.

Results

Linea 248       Indicate in the figure with arrows where the necrosis can be seen.

We have now indicated the inner area of the spheroid with arrows, which is mainly stained by eosin and lacks nuclei.

Line 250          The described observations are not as evident in the figure. This should be highlighted more.

We have now highlighted, using an arrow, an area where dead cells lack nuclei, serving as an example.

Line 270         Where in M&M do the authors talk about 2D cultivation?

In the M&M section dedicated to cell culture, the text reads:

“When cultured in 2D, hepatocytes were maintained in collagen-coated culture plates at a density of 10 million cells in a 150 mm petri dish for the indicated time.”

Line 286         Why were these markers considered? What do they highlight?

These markers are associated with extracellular vesicles, and their presence indicates that the preparations contain membrane-bound proteins associated with the process of vesicle formation and release, and are devoid of markers of intracellular vesicles. A reference is provided so that readers can verify the information.

Line 310-312  This statement needs to be supported by more data and considerations.

The text now reads:

“Remarkably, despite the co-culture of both primary hepatocytes and tumoral cells, the core of the spheroid consists of primary hepatocytes, as evidenced by Figures 5 and 6, which depict the arrangement of green fluorescent cells at the center of the spheroid, as well as Supplementary Figure 3, illustrating proliferation at the periphery of the section. Additionally, the presence of tumoral cells within the core cannot be overlooked, particularly in the SK-HEP1 co-culture, where proliferative cells infiltrate and disrupt the integrity of the inner core.”

Line 338-348  It is not specified how many replications were evaluated. How were the results processed? Where are described in details the chemical compounds?

The procedures were detailed in the flow cytometry section of the M&M, and the gating procedure is detailed in Supplemental Figure 5. The characteristics of the compounds now can be found in the discussion section.

The disaggregation of the spheroids, and flow cytometry:

“After a 20-minute incubation in Tryple (Gibco, ThermoFisher Scientific), spheroids were mechanically disaggregated with a 1000 µL pipette. Cells were washed with eBio-science™ Flow Cytometry Staining Buffer and suspended in a fluid containing a working concentration of SYTOX® Blue Dead Cell Stain (ThermoFisher Scientific). The suspension was immediately passed through a CytoFLEX flow cytometer (Beckman Coulter Ltd).”

In the result section, the process of result:

Therefore, we assessed the proportion of tumoral cells in the spheroids after the treatment, as well as quantified damaged cells using a dead cell stain that fluoresces at 450 nm (Sytox™, Thermofisher Scientifics). Both indicators revealed a reduction in tumoral cells in response to treatment, with Sorafenib showing a more pronounced effect. Notably, the dose employed for the cytotoxic drugs also inflicted damage on hepatocytes (Figure 8). Supplemental Figure 5 illustrates the gating strategy to differentiate tumoral from non-tumoral cells and vital from damaged cells, based on their staining with the vitality reporter.”

The legend of the Figure 8 also explains the procedure:

“Figure 8.  Percentage of damaged hepatocytes and tumoral cells in mono-cultures of primary hepatocytes spheroids (TMCRE), or mixed spheroids containing hepatocytes and different tumoral cells; HEPG2 (TMCRE_HEPG2), SKHEP-1 (TMCRE_SKHEP1), and HUH7 (TMCRE_HUH7). The results are percentages of stained cells according to flow cytometry analysis. In the graph, light green corresponds to vital hepatocytes (HP_LIVE), while dark green indicates damaged hepatocytes (HP_DEATH). Non-green bars represent tumoral cells, with light pink representing viable tumoral cells (TUMOR_LIVE), and the red segment of the bars indicating the percentage of damaged tumoral cells (TUMOR_DEATH). Data was obtained from a pool of 24 spheroids per condition. Treatment started at day 4, and spheroid collection was taken at day 14 after seeding.” 

Finally Supplemental Figure 5 describe the gating of hepatocytes and non-hepatocytes, as well as damaged vs non damaged cells:

“Supplemental Figure 5. Gatting of the tumoral cells and hepatocytes according to their fluorescence in the green channel. Afterwards, we separate vital from damaged cells by the fluorescence on the pacific blue channel, regarding the permeability to a vitality reporter. The examples correspond to hepatocyte spheroids, and to the mix spheroids with the tumoral cell SK-HEP1.”

Discussion and Conclusions

The conclusion chapter is poorly organized: the purpose of the work is not specified, the conclusions drawn by the authors are not sufficiently supported by the results of the work. Furthermore, not all results are discussed and compared with each other.

It would be useful to write an introductory sentence to contextualize the study.

An extensive review is necessary. It would be appropriate to rewrite the chapter

We have rewritten the discussion section, and for better understanding, we have subdivided in subchapters, dedicated to the different topics highlighted by the referees.

Figures

Figure 5         The caption needs to be more detailed

The caption has been rewritten.

“Figure 5. Tridimensional reconstruction of a series of confocal images taken from intact spheroids cultured for 7 days. Panel A shows the reconstruction of a mono-culture spheroid formed by primary hepatocytes (TMCRE), while Panels B, C, and D depict mixed spheroids containing tumoral cells HEPG2 (TMCRE_HEPG2), SKHEP-1 (TMCRE_SKHEP1), and HUH7 (TMCRE_HUH7), respectively. Tumoral cells, labelled before spheroid formation, may appear either reddish or uncoloured (see text for details), while hepatocytes exhibit green fluorescence.”

Round 2

Reviewer 4 Report

Comments and Suggestions for Authors

The authors provided balanced replied to my comments. Overall the manuscript is improved, thus can be published waiting for the eventual approval of the senior academic editor.

Comments on the Quality of English Language

done

Reviewer 5 Report

Comments and Suggestions for Authors

The authors reviewed the paper as indicated and it can be considered ready for publication.